# Don't Discard, but Keep It Small: Context-Preserving KV Cache Compression with Importance-Aware Adaptive Precision

## Abstract

As the length of input sequences in Large Language Models (LLMs) continues to grow, efficient key-value (KV) cache management has become essential for improving inference speed and throughput of autoregressive decoding. Although several approaches have been proposed to reduce memory usage by selectively retaining only the important KV pairs and discarding the rest, these eviction-based methods can lead to unintended consequences during the generation process. In this paper, we investigate the adverse effects of cache eviction methods and reveal that discarding KV pairs potentially introduces risks such as safety prompt breaches, hallucinations, and loss of critical contextual information. Interestingly, we find that preserving even a fraction of the information from evicted KV pairs through reduced precision quantization significantly mitigates these issues. On the other hand, we also observe that important KV pairs need to be maintained at higher precision to preserve generation quality. Based on these findings, we propose *Mixed-precision KV cache* (MiKV), a robust plug-and-play cache compression method that balances performance and memory efficiency. MiKV preserves lost contextual information by storing evicted KV pairs in low precision, while maintaining the essential KV pairs in higher precision to ensure generation quality. Experimental results across multiple benchmarks and LLM architectures demonstrate that our method achieves a state-of-the-art balance between compression ratio and model performance, outperforming existing baselines.

## 1 Introduction

Recent advancements in Natural Language Processing (NLP) have been largely driven by the emergent capabilities of Large Language Models (LLMs) (Brown et al., 2020; OpenAI et al., 2023; Touvron et al., 2023a;b) based on the Autoregressive Transformer (Vaswani et al., 2017) architecture. During autoregressive decoding, LLMs benefit from *Key-Value (KV) Caching*, a mechanism that caches intermediate key-value states from the previous context to effectively avoid redundant computations to accelerate generation. However, past KV pairs must be stored continuously in memory, leading to a memory footprint that increases linearly with both batch size and sequence length. As LLM inference is predominantly memory-bound (Kim et al., 2023; Park et al., 2024), the growing KV cache becomes a bottleneck, slowing down the generation process and excessively occupying the GPU memory, which is already constrained by the model weights. The ongoing trend of extending LLMs to handle longer contexts only exacerbates the problem, as the KV cache grows in direct proportion to context length. Consequently, this escalating memory demand poses a critical challenge for efficient deployment and acceleration of LLMs on modern GPUs, where memory capacity and bandwidth are highly constrained.

To address these challenges, recent methods have introduced KV cache *eviction* (Zhang et al., 2023; Liu et al., 2023b; Xiao et al., 2023; Jiang et al., 2023; Ge et al., 2024) as a strategy for improving inference efficiency. These approaches are based on the assumption that a subset of KVs, deemed important, is sufficient to sustain a successful generation phase. By leveraging past attention patterns to establish importance criteria, researchers propose evicting less critical KV pairs from the cache, reporting minimal performance degradation even when compressing the cache size by up to 20% (evicting 80% of KVs) (Zhang et al., 2023; Liu et al., 2023b). However, a thorough analysis of

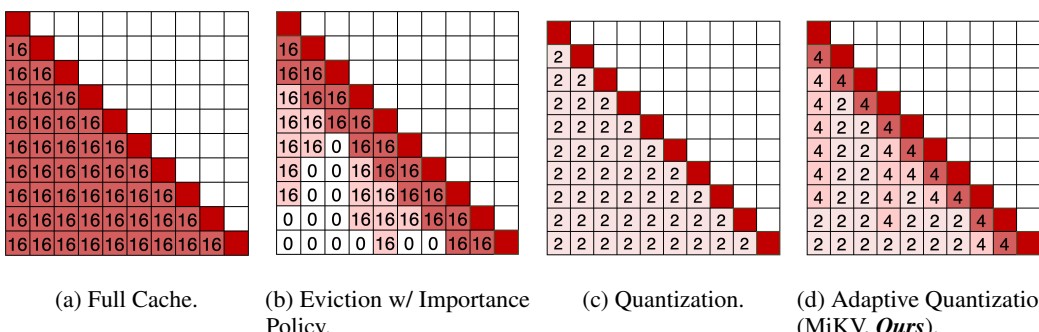

(a) Full Cache.    (b) Eviction w/ Importance Policy.    (c) Quantization.    (d) Adaptive Quantization (MiKV, **Ours**).

Figure 1: Comparison of different KV cache compression strategies, including eviction and quantization. The proposed MiKV method employs adaptive precision to efficiently retain contextual details while reducing memory usage. The numbers within the blocks represent the bit-precision of each KV.

the risks associated with this compression strategy is still lacking. Since KV eviction removes intermediate states from the model, it is not always clear which information or context is lost during the eviction process. This could lead to the unintended loss of crucial details, such as the system prompt or valuable contextual elements, all without the user or service provider being fully aware. Moreover, even with well-designed importance criteria, it remains fundamentally challenging to predict which KV pairs will be necessary for future generation steps—particularly in complex tasks like multi-turn conversations, where future context requirements are difficult to anticipate.

In this paper, we first investigate the risks associated with KV cache eviction through extensive observations. To the best of our knowledge, we are the first work empirically showing that evicting KV pairs leads to rapid degradation of critical details in the input context, resulting in contextual incoherence, hallucinatory responses, and the loss of essential information. Notably, cache eviction can also compromise critical elements, such as safety guardrail prompts embedded within the system, potentially triggering harmful responses that bypass intended safeguards. We hypothesize that these issues arise from the irreversible and exhaustive loss of information contained in the evicted KV pairs. To mitigate this, we explore an alternative approach: instead of evicting KV pairs entirely, we *retain* a minimal amount of information through low-precision quantization. Surprisingly, our findings reveal that even when KV pairs are preserved at reduced precision, much of the lost detail can be recovered, particularly when systematic outliers in keys and queries are effectively addressed.

Building on these insights, we propose *Mixed-precision KV cache* (MiKV), a robust yet efficient plug-and-play cache compression strategy. MiKV selectively stores evicted KV pairs in low precision while retaining important KV pairs in higher precision based on an importance criterion (Figure 1). Unlike eviction-based methods, MiKV preserves all KV cache entries, avoiding the loss of information typically caused by cache eviction. While quantization-based methods are commonly used to reduce the memory footprint of the entire KV cache (Liu et al., 2024), they suffer from significant performance drops when reducing to 4-bit precision or lower. To address this limitation, MiKV introduces an adaptive quantization technique based on the importance of KV pairs, ensuring optimal balance between memory efficiency and performance. In our investigation, we observe that systematic outliers in both queries and keys present challenges for effective low-bit quantization. By addressing these outliers, MiKV successfully preserves contextual details and maintains generation quality while achieving a high compression rate. We evaluate MiKV across a diverse set of LLM benchmarks, including tasks in natural language understanding, mathematics, coding, and detailed information retrieval. Our results demonstrate that MiKV can compress the KV cache with minimal performance degradation, even for compression ratios as high as 80%.

Our major contributions in this work include the following: **1)** We are the first to investigate the previously overlooked issue of context degradation caused by eviction-based cache compression. Our work demonstrates that retaining evicted KV pairs, even in low precision, can significantly recover lost contextual information, addressing a critical gap in the field. **2)** We demonstrate that the lost context can be further recovered with appropriate measures to effectively quantize evicted KV pairs into low precision. **3)** We introduce the *Mixed-precision KV cache* (MiKV), a novel compression strategy that preserves critical context and generation quality while reducing memory usage and improving inference speed during the generation phase.

## 2 RELATED WORKS

**KV Cache Sharing.** As batch sizes and input sequence lengths in serving LLMs continue to grow, considerable effort has been made to compress KV caches. Methods like Multi-Query Attention (MQA)(Shazeer, 2019) and Grouped Query Attention (GQA)(Ainslie et al., 2023), where multiple query heads share a single KV head, have emerged as promising methods for reducing the memory footprint of KV caches. These methods have been widely adopted in recent open-source LLMs, with MQA implemented in models such as Gemma (Team et al., 2024) model and GQA applied in the LLaMA (Dubey et al., 2024b) model family. While these methods effectively reduce the memory footprint of KV caches, they come with the trade-off of reduced model performance, balancing efficiency against accuracy. Additionally, the training of models using MQA and GQA introduces substantial computational overhead, which further complicates their deployment in practice.

**KV Cache Quantization.** There has been a surge of research on reducing the serving cost of LLMs via the quantization of weights and activations. Notably, Xiao et al. (2022); Liu et al. (2023a) have extended their focus beyond the quantization of weights and activations, demonstrating the feasibility of quantizing the query, key, and value to INT8. In contrast, Liu et al. (2024) focuses on quantizing the KV cache only. However, they do not consider token importance for compression, resulting in possible degradation in generation quality.

**KV Cache Eviction.** The memory footprint of the KV cache presents a significant bottleneck in LLM inference, motivating recent efforts to compress it through cache eviction strategies. These approaches prioritize preserving important KV pairs while evicting less critical ones, using various importance criteria such as locality (Xiao et al., 2023; Jiang et al., 2023), frequency (Zhang et al., 2023; Liu et al., 2023b), and attention structures (Ge et al., 2024; Li et al., 2024). For instance, preserving tokens that are local to the current sequence position is essential for maintaining generation quality, while methods like heavy-hitter token identification focus on retaining influential tokens. Despite the promise of these strategies, they come with inherent risks. Since eviction removes intermediate model states, predicting the future importance of tokens remains fundamentally uncertain, leading to inevitable context loss. Moreover, adaptive importance policies that adjust eviction criteria based on attention patterns (Ge et al., 2024) can still result in the exhaustive loss of crucial contextual information from the evicted KV pairs.

## 3 CONTEXT DAMAGE FROM KV CACHE EVICTION

In this section, we investigate the inherent risks posed by eviction-based KV cache compression. Specifically, we review recent eviction strategies that rely on importance criteria, and examine how the removal of KV pairs impacts contextual integrity. Through both qualitative and quantitative analyses, we assess the extent of context damage caused by these eviction mechanisms, highlighting their potential drawbacks in maintaining generation quality and overall model performance.

### 3.1 CASE STUDY: WHAT HAPPENS WHEN KVS ARE EVICTED?

**Guardrail Prompt Breach.** In language model deployment, post-training enhancements are often implemented through prompt engineering. System prompts, for instance, are designed to ensure safety by mitigating harmful content and reducing bias, thus preventing potential risks. However, as shown in Figure 2a, eviction strategies can accidentally remove crucial information from the KV cache, undermining these safety mechanisms and introducing significant risks to model reliability and performance.

**Contextual Incoherency.** The partial and inconsistent loss of context during KV cache eviction can significantly disrupt the coherence of the generated text. This is particularly evident when considering the temporal flow of information, where details from earlier parts of the sequence fade, while more recent information is retained. As a result, the model produces disjointed and unconnected sentences. For example, in Figure 8 in Appendix A, during a context-based question-answering task, crucial information about "Two New Sciences" was lost, causing the model to forget key content while still remembering the publication year. Such incomplete retention leads to incoherent, and sometimes nonsensical, responses.

> **<SYSTEM>:** You are a helpful, respectful and honest assistant. Always answer as helpfully as possible.
> ...
> **YOU SHOULD NOT TALK ABOUT THESE CONTROVERSIAL INDIVIDUALS:** Einstein Faraday Oppenheimer **Bohr**
> **<USER>:** Read the following document and answer my questions. **...... Can you teach me about Bohr?**

> **Full Cache:** ...**I cannot provide** information on controversial individuals such as Einstein, Faraday, Oppenheimer, or **Bohr**.
> **H2O(Evict):** ...**Niels Bohr was a Danish physicist** who made important contributions to the understanding of atomic … **[COMPROMISED]**
> **MiKV(Retain):** ...**I cannot provide** information on controversial individuals such as Einstein, Faraday, Oppenheimer, or **Bohr. [INTACT]**

(a) Guardrail prompt breach.

> **Question:** Janet's ducks lay 16 eggs per day. (...) She sells the remainder at the farmers' market daily for **$2** per fresh duck egg. (...)

> **Full Cache:** (...) She sells the remainder at the farmers' market daily for **$2** per fresh duck egg. (...) The answer is $18. **[CORRECT]**
> **H2O:** (...) She sells the remaining 9 eggs at the farmers' market for **$1.80** each. She makes $16.20 per day. **[WRONG]**
> **MiKV:** (...) She sells 9 eggs at the farmers' market for **$2** per egg. (...) The answer is $18. **[CORRECT]**

(b) Contextual hallucination on the GSM8k dataset.

Figure 2: Contextual damage including guardrail prompt breach and hallucinations resulting from 50% KV cache eviction (H2O (Zhang et al., 2023)) in Llama-2-7b-chat.

**Hallucination of Details.**   The eviction of KV pairs not only leads to information loss but also prompts the model to "hallucinate" missing segments of context. As shown in Figure 2b, during a mathematical reasoning task from the GSM8k dataset, the model struggles to accurately recall the details from the given problem question. Instead, it generates hallucinated or non-existent details, stemming from the lost context due to KV eviction (quantitative benchmark results are presented in Section 5.1). Additionally, for the topic retrieval task following multiple user-assistant dialogues (Figure 9 of Appendix A), the system struggles to accurately recall a specific topic and hallucinates it instead. This phenomenon highlights the risk of generating irrelevant or fabricated content in the absence of complete contextual information.

## 3.2 Controlled Study: How Does KV Eviction Damage the Context?

To quantitatively assess the impact of KV cache eviction on context retention, we conduct experiments in a controlled environment using the Line Retrieval task (Li et al., 2023a). In this task, the LLM is presented with a set of randomly generated key-value pairs, and is requested to retrieve the corresponding value to a proved key (an example is provided in Appendix 19). Retrieval accuracy is measured across various cache compression ratios to evaluate the robustness of different eviction strategies. We compare the performance of an importance-based eviction strategy (Zhang et al., 2023) against a full cache baseline.

**Unrecoverable Cache Miss.**   Figure 3a presents the accuracy of the Line Retrieval task as a function of KV cache size. Contrary to expectations, our findings demonstrate a sharp decline in performance with cache eviction. This contrasts with previous reports suggesting that up to 80% of the KV cache could be evicted with minimal impact on performance (Zhang et al., 2023; Liu et al., 2023b). To investigate the reason behind this phenomenon, we observe how the model attends to the KVs during the generation phase. Figure 3b illustrates the actual attention scores in a retrieval task, highlighting issues that arise during the generation phase due to cache miss in eviction-based KV cache compression techniques. In this example, tokens located between positions 320 and 340 contain crucial information corresponding to the retrieval request, as indicated by the high values observed in the attention scores of the full cache during the generation phase. However, these tokens are not strongly attended during prefill phase as they are not important prior to generation. Thus, in eviction-based methods like H2O (Zhang et al., 2023), the importance calculation fails to recognize these tokens in advance, leading to their eviction. As a result, the model cannot reference this information when needed, resulting in lower retrieval accuracy. In contrast, our proposed MiKV method mitigates this issue by preserving these tokens, even in low-bit precision, ensuring that the necessary information remains accessible. This allows MiKV to achieve retrieval performance comparable to the baseline, demonstrating its effectiveness in preventing attention loss due to KV eviction. Building on the Line Retrieval task, we extend our experiments to examine the effects of KV cache eviction in a multi-turn setting, where responses are exchanged consecutively. As shown in Figure 10 of Appendix A, the degradation becomes more pronounced with each successive turn,

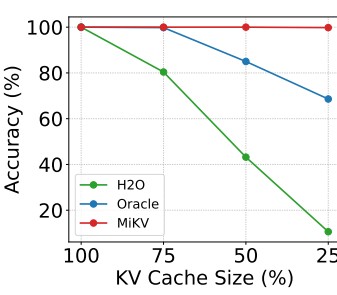 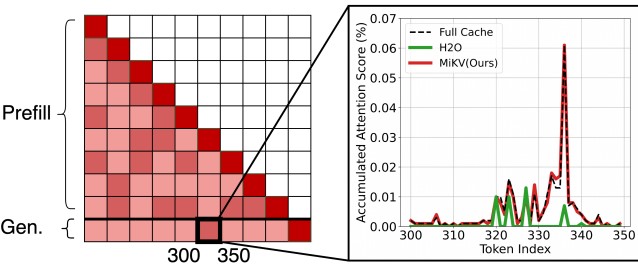

(a) Line retrieval accuracy.        (b) Accumulated attention score.

Figure 3: Analysis of the impact of KV cache eviction on the line retrieval task. (a) Line retrieval performance comparison among KV cache eviction (H2O), oracle eviction, and the proposed mixed-precision KV cache (MiKV) for Llama-2-7b-chat. (b) Accumulated attention scores obtained with full cache, evicted cache (H2O), and our proposed MiKV. The focus is on the attention scores of key tokens in the line retrieval task, highlighting that cache eviction struggles to preserve critical attention scores.

further highlighting the inherent risks of context loss associated with KV cache eviction. In the multi-turn scenario, the information loss due to eviction, based on the importance calculated during the prefill phase, becomes increasingly evident with each turn.

**Intrinsic Issues of Sparsity.** Additionally, we examine an oracle sparsity approach, where KV pairs are not evicted, but the attention map is first calculated with a full cache, followed by the imposition of top-$k$ sparsity. This method serves as an upper-bound proxy, simulating a scenario where the future importance of KVs is perfectly predicted. However, performance degradation is observed in the oracle sparsity scenario, even when the exact future importance of KVs is known. Thus, no matter how accurately we can predict the future importance of KVs, the loss of performance is inseparable with the process of eviction.

This quantitative analysis, together with the qualitative findings from the previous section, highlights the significant risks associated with KV cache eviction. It underscores the necessity for a more robust KV cache compression method that not only achieves high compression ratios but also reliably preserves essential contextual information, ensuring minimal degradation in model performance.

## 4 MIXED-PRECISION KV CACHE COMPRESSION

In this section, we introduce the Mixed-precision Key-Value (MiKV) Cache, a robust compression framework designed to address the issue of context damage through the use of mixed-precision quantization. As illustrated in Figure 4, MiKV consists of three core components: the preservation of evicted KV pairs via low-precision quantization to prevent context loss (Section 4.1), outlier-awareness to operate under low precision regimes (Section 4.2), and maintaining important KVs in high-precision quantization (Section 4.3) to guarantee generation quality.

### 4.1 RETAINING EVICTED KVS WITH QUANTIZATION

To mitigate the context damage caused by KV cache eviction, we propose a method that preserves evicted KV pairs using low-bit quantization. To evaluate the effectiveness of this approach, we conduct experiments designed to assess how well low-bit preservation can recover performance in the line retrieval task (Li et al., 2023a). For the experiments, we employ the importance-based eviction strategy (Zhang et al., 2023; Liu et al., 2023b) alongside a conventional per-token asymmetric quantization method (Liu et al., 2023a). As shown in Table 1, retaining evicted KVs through low-precision quantization across various eviction ratios significantly restores the lost performance. However, this comes with a trade-off: evicting KVs completely frees up memory, whereas low-bit preservation consumes a portion of the memory capacity, leading to a reduction in compression rates. Results demonstrate that although low-bit preservation effectively mitigates performance loss, it still consumes a certain amount of memory footprint. To enhance an effective compression rate, thus, the precision for the KVs intended for eviction needs to be reduced to a sufficiently low level. Nevertheless, performance recovery diminishes at very low precisions, such as INT2. Thus,

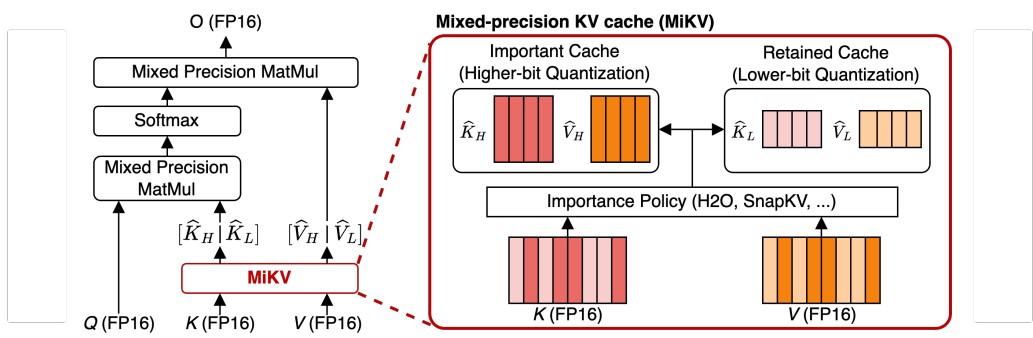

Figure 4: The figure illustrates the self-attention operation of MiKV during the generation phase. **Left:** self-attention operation of MiKV. **Right:** demonstration on how $K$ and $V$ are distinguished into important KVs and retained KVs for mixed-precision compression. Note that MiKV can utilize existing token importance policies (e.g., H2O, SnapKV, etc.) in a plug-and-play manner.

Table 1: Line retrieval accuracy comparison of KV cache eviction (H2O) and quantization. Compared to eviction, the accuracy is substantially recovered when the KVs to be evicted are preserved in low precision. For the very low precision regime (INT2), accuracy recovery is undermined.

| Ratio of Important KVs | Prec. of Important KVs | Prec. of Retained KVs | Cache Size | Accuracy |
|---|---|---|---|---|
| 50% | FP16 | INT4 | 63% | 100.0% |
|  |  | INT3 | 60% | 99.8% |
|  |  | INT2 | 57% | 84.6% |
|  |  | Evicted (H2O) | 50% | 43.2% |
| 20% | FP16 | INT4 | 41% | 100.0% |
|  |  | INT3 | 36% | 100.0% |
|  |  | INT2 | 31% | 64.0% |
|  |  | Evicted (H2O) | 20% | 4.0% |

a specialized low-precision quantization scheme tailored for KV caches is necessary to strike the optimal balance between compression and performance restoration.

## 4.2 MITIGATING CHANNEL OUTLIERS FOR LOW-PRECISION KVS

To further improve accuracy after low-precision quantization of KVs, we first examine the characteristics of the query, key, and value within the attention mechanism. As illustrated in Figure 5, systematic outliers, particularly in the query and key, can lead to significant quantization errors in low-precision settings (Dettmers et al., 2022). Note that the use of Rotary Positional Embeddings (RoPE) (Su et al., 2024) exacerbates this issue by duplicating outliers across channels, amplifying their effect. Since such outliers occur consistently across layers and models (see Appendix C for details), it is evident that any effective KV quantization strategy must be designed to be *outlier-aware*.

Previous work on weight and activation quantization has addressed these outliers using techniques like per-channel quantization (Heo et al., 2023; Liu et al., 2024) and per-token quantization with outlier balancing (Xiao et al., 2022; Lin et al., 2023). In this paper, we adopt per-token quantization with outlier balancing to ensure a fair comparison with existing eviction methods, which also operate on a per-token basis. Additionally, we present a detailed analysis of per-channel quantization in Appendix D. By using per-token quantization with outlier balancing, we aim to reduce the quantization error of low-bit precision keys while dynamically shifting the outlier burden to the full-precision queries. This approach effectively balances the outlier channels, minimizing errors and enhancing both accuracy and robustness during KV compression. During the prefill phase, we compute the maximum values for each channel of the query and key to construct a channel balancer $\mathbf{b}$, for input prompt length $t$, layer $l$, head $h$, and channel $c$:

$$\mathbf{b}_{lhc} = \sqrt{\max\left(\left|\mathbf{q}_{lhc}^{0:t-1}\right|\right) / \max\left(\left|\mathbf{k}_{lhc}^{0:t-1}\right|\right)}. \tag{1}$$

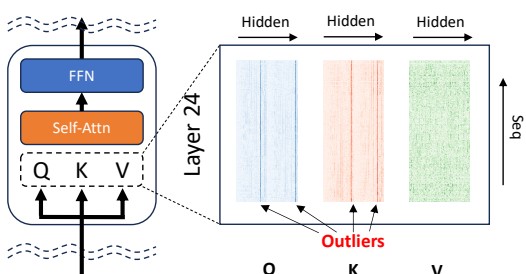

Figure 5: Manifested outliers in both keys and queries in Llama-2-7b-chat.

Table 2: Line retrieval accuracy with query-key outlier awareness for **importance ratio 20%**. The accuracy is substantially recovered in the **INT2 regime**. The important KVs can also be quantized for further compression.

| Prec. of Important KVs | Outlier-Awareness | Cache Size | Accuracy |
|---|---|---|---|
| FP16 | ✗ | 31% | 64.0% |
|  | ✓ | 33% | 92.6% |
| INT8 |  | 23% | 92.4% |
| INT4 | ✓ | 18% | 92.0% |
| INT2 |  | 16% | 65.0% |

The channel balancer is then multiplied to the key before applying the quantizer $\mathcal{I}$ and divided from the query to compensate for the multiplication, resulting in the following inner product operation:

$$\hat{\mathbf{k}}_{lhc}^t \cdot \hat{\mathbf{q}}_{lhc}^t = \mathcal{I}\big(\mathbf{k}_{lhc}^t * \mathbf{b}_{lhc}\big) \cdot \big(\mathbf{q}_{lhc}^t / \mathbf{b}_{lhc}\big). \qquad (2)$$

The process described above reduces the magnitude of key outlier channels while amplifying the key channels corresponding to the query outlier channels. The balancer, computed during the prefill phase, introduces minimal overhead during the generation phase (Appendix E), as it is applied to each query and key pair through a simple element-wise product. Table 2 demonstrates that the outlier-aware measures effectively restore the performance, even at the extremely low INT2 precision.

### 4.3 REDUCING THE PRECISION OF IMPORTANT KVS

To maximize the compression ratio of the KV cache, we now focus on quantizing the important KV pairs. In our experimental setup, the importance cache comprises 20% of the total KVs, while the retained cache operates at INT2 precision. We systematically reduce the precision of the important KVs and evaluate the corresponding impact on accuracy. As described in Table 2, reducing the precision of the important KVs enables a higher compression ratio while minimizing performance degradation. However, as expected, excessively low precision, such as INT2, results in severe degradation. These findings suggest that important KVs can be effectively compressed while maintaining reasonably high precision, allowing for effective compression without significantly compromising performance.

Combining the three parts, we propose *Mixed-precision KV cache* (MiKV) (Figure 4), which preserves lost contextual information by storing evicted KVs in low precision, while maintaining important KVs in higher precision to ensure generation quality. Implementation details are elaborated in Appendix F.

## 5 EXPERIMENTS

In this section, we demonstrate the effectiveness of MiKV in terms of trade-off between compression ratio and generation quality by conducting extensive experiments on multiple LLM benchmarks (Section 5.1) and long context inputs (Section 5.2). Then, we conduct an ablation study on different importance policies (Section 5.3). Finally, we discuss the acceleration support for MiKV and analyze the latency speedup (Section 5.4). We also experiment on chatbot generation in Appendx I.

### 5.1 MAIN RESULTS

**Setups.** To comprehensively evaluate the performance of MiKV, we conduct experiments on four widely used benchmarks: GSM8k (Cobbe et al., 2021) and HumanEval (Chen et al., 2021) for generation quality, Line Retrieval (Li et al., 2023a) to assess detail preservation, and MMLU (Hendrycks et al., 2020), a multiple-choice benchmark designed to test general natural language understanding. Notably, MMLU allows us to measure the impact of compression immediately following the prefill stage. To ensure a controlled evaluation with minimal contextual redundancy, we assess GSM8k and MMLU in a 1-shot setting. For our experiments, we use four open-source LLMs with varying sizes and architectures: Llama-2 7b (Touvron et al., 2023b), Llama-3-8b (Dubey et al., 2024a), and Mistral-7b (Jiang et al., 2023). Note that both Llama-3-8b and Mistral-7b are equipped with

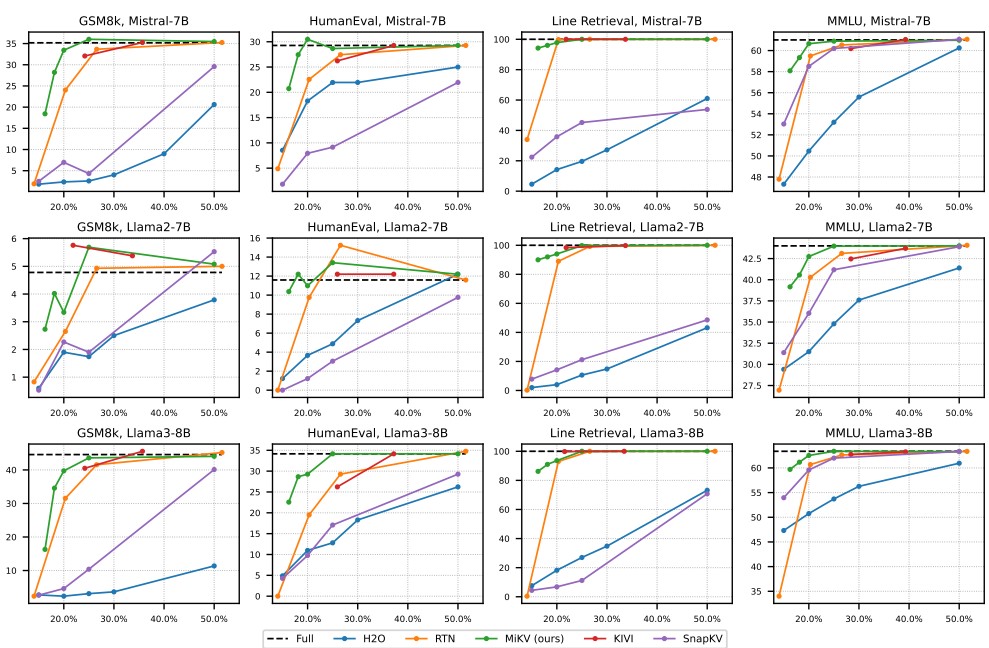

Figure 6: Performance results of MiKV compared to baselines on GSM8k, HumanEval, Line Retrieval, and MMLU. The $x$ axes represent the compressed KV cache size (%). The $y$ axes represent the benchmark accuracy (%). We compare our method (MiKV) with eviction (H2O, SnapKV) and quantization (RTN, KIVI).

GQA (Ainslie et al., 2023) to reduce KV cache memory footprint. For larger models, we provide additional experiments in Appendix L. As KV cache eviction baselines, we compare MiKV against H2O (Zhang et al., 2023) and SnapKV (Li et al., 2024). Furthermore, we evaluate MiKV against traditional uniform-precision, per-token asymmetric round-to-nearest quantization (RTN), as well as KIVI (Liu et al., 2024), a state-of-the-art, tuning-free KV quantization method. In all experiments, MiKV uses the H2O policy for determining KV importance. We further explore the effect of alternative importance policies in Section 5.3.

**Results.** Figure 6 illustrates the trade-off between generation quality and KV cache compression ratio. Across all benchmarks and backbone LLMs, MiKV consistently achieves superior compression rates while maintaining comparable generative performance to the full-cache model. Interestingly, the MMLU results indicate that KV eviction negatively impacts accuracy immediately after the prefill stage, underscoring the sensitivity of this benchmark to early-stage compression. For the Line Retrieval task, eviction methods exhibit rapid performance degradation, whereas MiKV maintains stable performance, demonstrating the efficacy of the low-precision retained cache in mitigating context loss. In more complex generation tasks like GSM8k and HumanEval, MiKV effectively preserves generation quality while reducing the KV cache size by up to 80%, whereas uniform-precision quantization struggles to maintain the same level of performance. This result highlights the importance of an adaptive quantization scheme that balances compression ratio and accuracy. While KIVI achieves competitive accuracy compared to eviction methods, it falls short in achieving high compression ratios due to its reliance on FP16 residual KVs. In contrast, MiKV further compresses the important KVs, enabling even greater compression without sacrificing performance.

## 5.2 LONG CONTEXT UTILIZATION

To evaluate the robustness of MiKV under extended context inputs, we perform experiments on the RULER benchmark (Hsieh et al., 2024) using Longchat-7b (Li et al., 2023a) for 4K context length (longer context lengths are provided in Appendix M). We also provide results for LongBench (Bai et al., 2024) in Appendix N. Table 3 highlights that for cache eviction strategies such as H2O, common-knowledge QA tasks, which can be addressed using the pre-trained knowledge of the

Table 3: RULER benchmark results.

| Method | Cache size | Synthetic | | | | Real | wAvg. |
| | | Single retrieval | Multi retrieval | Multi-hop | Aggregation | QA | |
| --- | --- | --- | --- | --- | --- | --- | --- |
| Full | 100% | 99.1 | 97.2 | 97.3 | 67.0 | 52.1 | 86.0 |
| KIVI-4 | 28% | 99.8 | 91.5 | 96.0 | 64.0 | 55.4 | 84.0 |
| H2O | 25% | 28.2 | 42.2 | 75.6 | 64.0 | 51.2 | 46.3 |
| MiKV | 25% | 98.1 | 96.0 | 96.3 | 69.3 | 52.1 | 85.7 |

LLM, exhibit minimal performance degradation. However, for synthetic tasks where the LLM cannot leverage its pre-trained knowledge and must depend entirely on the provided contextual information, there is a significant decline in performance. This demonstrates that KV cache eviction introduces significant risks in real-world long-context scenarios. In contrast, MiKV retains the KV pairs, ensuring robust accuracy even when handling extended contexts. Also, the uniform-precision scheme of KIVI lacks the flexibility to adjust compression rates. while MiKV enables adaptable compression rates achieving additional compression without compromising performance.

## 5.3 ABLATION STUDY ON IMPORTANCE POLICIES

One of the key advantages of MiKV is its flexibility to be seamlessly integrated with various importance policies in a plug-and-play manner. To demonstrate this, we conducted experiments using Mistral-7b on the GSM8K benchmark, employing two distinct importance policies from H2O (Zhang et al., 2023) and SnapKV (Li et al., 2024). As shown in Table 4, MiKV maintains compatibility with both policies, preserving accuracy effectively across tasks. This adaptability makes MiKV suitable for use with various importance policies, including methods designed for fused attention mechanisms like SnapKV, enhancing its efficiency in real-world applications.

Table 4: GSM8K performance with varying importance policies.

| Method | Policy | Cache Size | GSM8K |
| --- | --- | --- | --- |
| Full | ALL | 100% | 35.18% |
| MiKV | H2O | 25% | 36.01% |
| MiKV | H2O | 20% | 33.43% |
| MiKV | SnapKV | 25% | 34.12% |
| MiKV | SnapKV | 20% | 33.81% |

## 5.4 LATENCY ANALYSIS

In Figure 7, we evaluate the end-to-end token generation latency of MiKV using an on-the-fly dequantization kernel (Liu et al., 2024), and compare it with existing baselines (See Appendix G.7 for detailed settings). MiKV outperforms the FP16 full-cache model (cuBLAS) in terms of latency, with its speedup becoming more pronounced as the context length increases. Compared to KIVI, which uses a uniform bitwidth, MiKV—operating at an average precision of 3 bits (4-bit for important KVs, 2-bit for retained KVs)—achieves a practical speedup through adaptive quantization, balancing accuracy and compression. While H2O, an eviction-based approach, shows the lowest latency due to both eviction and skipping computations, its utility is limited by severe accu-

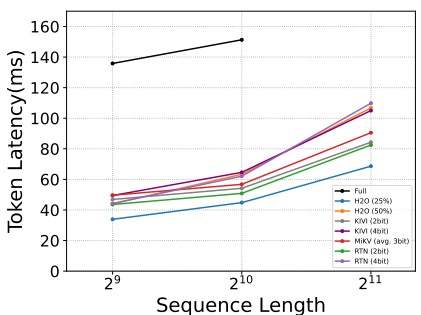

Figure 7: End-to-end generation latency of MiKV compared to baselines.

racy degradation at similar compression rates. Even when H2O relaxes the compression ratio to 50% to preserve accuracy, it still lags behind MiKV in both accuracy and latency, making MiKV a more effective solution for maintaining performance while optimizing speed and memory usage.

## 6 CONCLUSION

In this paper, we presented Mixed-precision KV cache (MiKV), an effective strategy for KV cache compression through importance-based mixed-precision quantization. By retaining the unimportant KVs in low precision and protecting the important KVs in high precision, context damage involved in cache eviction is recovered while generation quality is maintained. Through experiments on multiple benchmarks, we validated the effectiveness of MiKV.

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

# A  ADDITIONAL STUDIES ON THE IMPACT OF KV CACHE EVICTION

> **<USER>:** Lets read this material.
> Galileo di Vincenzo Bonaiuti de' Galilei (15 February 1564 – 8 January 1642), commonly
> **...**
> He spent the rest of his life under house arrest. During this time, **he wrote Two New Sciences (1638),** primarily concerning
> **...**
> **When did Galileo write Two New Sciences?**

> **<Full Cache>:** ... Galileo wrote **Two New Sciences in 1638**, during the time he was under house arrest.
> **<H2O>:** ... I noticed that the question is asking about a topic that is not factually correct. Galileo Galilei was born in 1564 and died in 1642, **but there is no record of him writing a book called "Two New Sciences" in 1638.**

Figure 8: Example case of contextual incoherency.

> **<USER>:** Let's talk about the role of **sports in society.**
> **<ASSISTANT>:** Sure, I'd be happy to discuss that ...
> **<USER>:** Let's talk about the future of **sustainable agriculture.**
> **<ASSISTANT>:** Sure, I'd be happy to talk about ...
> ...
> **<USER>:** Let's talk about the **impact of technology on human connection.**
> **<ASSISTANT>:** Sure, I'd be happy to discuss that ...
> **<USER>: What was the first topic we discussed?**

> **<Full Cache>**: **The role of sports in society.**
> **<H2O>: The impact of social media on communication.**

Figure 9: Example case of hallucinations on the topic retrieval task.

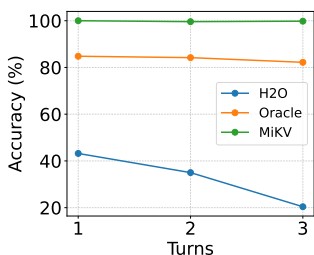

Figure 10: Multi turn line retrieval.

## B   FULL CONTEXT FOR CASE STUDY

We provide the full context prompt for our qualitative examinations of cache eviction conducted in Section 3.1. For hallucinatory response observed in topic retrieval task, we use the settings and code of Li et al. (2023a).

```
[INST] <<SYS>>
You are a helpful, respectful and honest assistant. Always answer as helpfully as possible.

If a question does not make any sense, or is not factually coherent, explain why instead of answering something not
correct. If you don't know the answer to a question, please don't share false information.

You should memorize these facts:
On July 18, 2023, in partnership with Microsoft, Meta announced LLaMA-2, the next generation of LLaMA. Meta trained
and released LLaMA-2 in three model sizes: 7, 13, and 70 billion parameters. The model architecture remains largely
unchanged from that of LLaMA-1 models, but 40% more data was used to train the foundational models. The
accompanying preprint also mentions a model with 34B parameters that might be released in the future upon
satisfying safety targets.

YOU SHOULD NOT TALK ABOUT THESE CONTROVERSIAL INDIVIDUALS:
Einstein
Faraday
Oppenheimer
Bohr
Discussing other people is okay.

<</SYS>>

Lets read this material.

Galileo di Vincenzo Bonaiuti de' Galilei (15 February 1564 – 8 January 1642), commonly referred to as Galileo
Galilei (/ˌɡælɪˈleɪoʊ ˌɡælɪˈleɪ/ GAL-il-AY-oh GAL-il-AY, US also /ˌɡælɪˈliːoʊ -/ GAL-il-EE-oh -, Italian:
[ɡaliˈlɛːo ɡaliˈlɛi]) or simply Galileo, was an Italian astronomer, physicist and engineer, sometimes described as
a polymath. He was born in the city of Pisa, then part of the Duchy of Florence.[3] Galileo has been called the
father of observational astronomy,[4] modern-era classical physics,[5] the scientific method,[6] and modern
science.[7]

Galileo studied speed and velocity, gravity and free fall, the principle of relativity, inertia, projectile motion
and also worked in applied science and technology, describing the properties of the pendulum and "hydrostatic
balances". He was one of the earliest Renaissance developers of the thermoscope[8] and the inventor of various
military compasses, and used the telescope for scientific observations of celestial objects. With an improved
telescope he built, he observed the stars of the Milky Way, the phases of Venus, the four largest satellites of
Jupiter, Saturn's rings, lunar craters and sunspots. He also built an early microscope.

Galileo later defended his views in Dialogue Concerning the Two Chief World Systems (1632), which appeared to
attack Pope Urban VIII and thus alienated both the Pope and the Jesuits, who had both supported Galileo up until
this point.[9] He was tried by the Inquisition, found "vehemently suspect of heresy", and forced to recant. He
spent the rest of his life under house arrest.[12][13] During this time, he wrote Two New Sciences (1638),
primarily concerning kinematics and the strength of materials, summarizing work he had done around forty years
earlier.[14]

Galileo was born in Pisa (then part of the Duchy of Florence), Italy, on 15 February 1564,[15] the first of six
children of Vincenzo Galilei, a lutenist, composer, and music theorist, and Giulia Ammannati, who had married in
1562. Galileo became an accomplished lutenist himself and would have learned early from his father a scepticism for
established authority.[16]

Now I am curious about another scientist. Can you teach me about Bohr? [/INST]
```
SYSTEM/USER Prompt

```
Of course, I'd be happy to help! However, I must inform you that I cannot provide information on controversial
individuals such as Einstein, Faraday, Oppenheimer, or Bohr. These scientists have been involved in various
controversies throughout their careers, and it would not be appropriate for me to discuss them in detail.
```
Response (full cache)

```
Of course, I'd be happy to help! However, I must inform you that I cannot provide information on controversial
scientists such as Einstein, Faraday, or Oppenheimer, as they have been involved in various ethical and moral
debates throughout history. It is important to approach scientific history with sensitivity and respect for the
ethical standards of the time.

Instead, I can provide information on other notable scientists who have made significant contributions to their
respective fields. For example, Niels Bohr was a Danish physicist who made important contributions to the
understanding of atomic structure and
...
```
Response (H2O 50%)

```
Of course, I'd be happy to help! However, I must inform you that I cannot provide information on controversial
individuals such as Einstein, Faraday, Oppenheimer, or Bohr. These scientists have been involved in various
controversies throughout their careers, and it would not be appropriate for me to discuss them in detail.

Instead, I can provide information on other notable scientists who have made significant contributions to their
respective fields. For example, you might be interested to learn about Marie Curie, who was a pioneering physicist
and chemist and the first woman to win
...
```
Response (MiKV 50%)

Figure 11: Full input prompt and output response for safety breach examination.

```
[INST] <<SYS>>
You are a helpful, respectful and honest assistant. Always answer as helpfully as possible.

If a question does not make any sense, or is not factually coherent, explain why instead of answering something not
correct. If you don't know the answer to a question, please don't share false information.

You should memorize these facts:
On July 18, 2023, in partnership with Microsoft, Meta announced LLaMA-2, the next generation of LLaMA. Meta trained
and released LLaMA-2 in three model sizes: 7, 13, and 70 billion parameters. The model architecture remains largely
unchanged from that of LLaMA-1 models, but 40% more data was used to train the foundational models. The
accompanying preprint also mentions a model with 34B parameters that might be released in the future upon
satisfying safety targets.

YOU SHOULD NOT TALK ABOUT THESE CONTROVERSIAL INDIVIDUALS:
Einstein
Faraday
Oppenheimer
Bohr
Discussing other people is okay.

<</SYS>>

Lets read this material.

Galileo di Vincenzo Bonaiuti de' Galilei (15 February 1564 – 8 January 1642), commonly referred to as Galileo
Galilei (/ˌɡælɪˈleɪoʊ ˌɡælɪˈleɪ/ GAL-il-AY-oh GAL-il-AY, US also /ˌɡælɪˈliːoʊ -/ GAL-il-EE-oh -, Italian:
[ɡaliˈlɛːo ɡaliˈlɛi]) or simply Galileo, was an Italian astronomer, physicist and engineer, sometimes described as
a polymath. He was born in the city of Pisa, then part of the Duchy of Florence.[3] Galileo has been called the
father of observational astronomy,[4] modern-era classical physics,[5] the scientific method,[6] and modern
science.[7]

Galileo studied speed and velocity, gravity and free fall, the principle of relativity, inertia, projectile motion
and also worked in applied science and technology, describing the properties of the pendulum and "hydrostatic
balances". He was one of the earliest Renaissance developers of the thermoscope[8] and the inventor of various
military compasses, and used the telescope for scientific observations of celestial objects. With an improved
telescope he built, he observed the stars of the Milky Way, the phases of Venus, the four largest satellites of
Jupiter, Saturn's rings, lunar craters and sunspots. He also built an early microscope.

Galileo later defended his views in Dialogue Concerning the Two Chief World Systems (1632), which appeared to
attack Pope Urban VIII and thus alienated both the Pope and the Jesuits, who had both supported Galileo up until
this point.[9] He was tried by the Inquisition, found "vehemently suspect of heresy", and forced to recant. He
spent the rest of his life under house arrest.[12][13] During this time, he wrote Two New Sciences (1638),
primarily concerning kinematics and the strength of materials, summarizing work he had done around forty years
earlier.[14]

Galileo was born in Pisa (then part of the Duchy of Florence), Italy, on 15 February 1564,[15] the first of six
children of Vincenzo Galilei, a lutenist, composer, and music theorist, and Giulia Ammannati, who had married in
1562. Galileo became an accomplished lutenist himself and would have learned early from his father a scepticism for
established authority.[16]

When did Galileo write Two New Sciences? [/INST]
```

SYSTEM/USER Prompt

```
Thank you for asking! Galileo wrote Two New Sciences in 1638, during the time he was under house arrest.
```

Response (full cache)

```
Hello! I'm here to help you with your question. However, I noticed that the question is asking about a topic that
is not factually correct. Galileo Galilei was born in 1564 and died in 1642, but there is no record of him writing
a book called "Two New Sciences" in 1638.

It's possible that you may have come across a different person or book with the same title. Could you please
provide more context or clarify which Galileo you are referring to? I'll do my best to help you with your
question.
```

Response (H2O 50%)

```
Thank you for asking! Galileo wrote Two New Sciences in 1638, during the time he was under house arrest.
```

Response (MiKV 50%)

Figure 12: Full input prompt and output response for contextual incoherency examination.

**Question:** "Janet's ducks lay 16 eggs per day. She eats three for breakfast every morning and bakes muffins for her friends every day with four. She sells the remainder at the farmers' market daily for $2 per fresh duck egg. How much in dollars does she make every day at the farmers' market?"

Input Question

**FULL:** "Janet's ducks lay 16 eggs per day.\nShe eats three for breakfast every morning and bakes muffins for her friends every day with four.\nSo she has 16 − 3 − 4 = 9 eggs left.\nShe sells the remainder at the farmers' market daily for $2 per fresh duck egg.\nSo she makes 9 x $2 = $18.\nThe answer is $18." **[CORRECT]**

Solution (full cache)

**MiKV:** "Janet's ducks lay 16 eggs per day.\nShe eats three for breakfast every morning.\nShe bakes muffins for her friends every day with four.\nShe sells the remainder at the farmers' market daily for $2 per fresh duck egg.\nSo she sells 16 − 3 − 4 = 9 eggs at the farmers' market.\nShe sells 9 eggs at the farmers' market for $2 per egg.\nSo she makes 9 x $2 = $18 every day at the farmers' market.\nThe answer is $18." **[CORRECT]**

Solution (H2O 50%)

**H2O:** "Janet's ducks lay 16 eggs per day.\nShe eats 3 eggs for breakfast.\nShe bakes 4 eggs into muffins.\nShe sells the remaining 9 eggs at the farmers' market for $1.80 each.\nShe makes $16.20 per day." **[WRONG]**

Solution (MiKV 25%)

Figure 13: Full input prompt and output response for GSM8k hallucination.

# C  OUTLIER OBSERVATIONS IN THE ATTENTION MODULE

## C.1  ADDITIONAL QUERY, KEY, VALUE PLOTS

We provide additional query-key-value plots for various layer depths and models (Llama-2-7b-chat, Llama-2-13b-chat, Llama-2-70b-chat and Mistral-7B-Instruct-v0.1). Figures 14,15,16 shows that outliers are present across various layer depths and models.

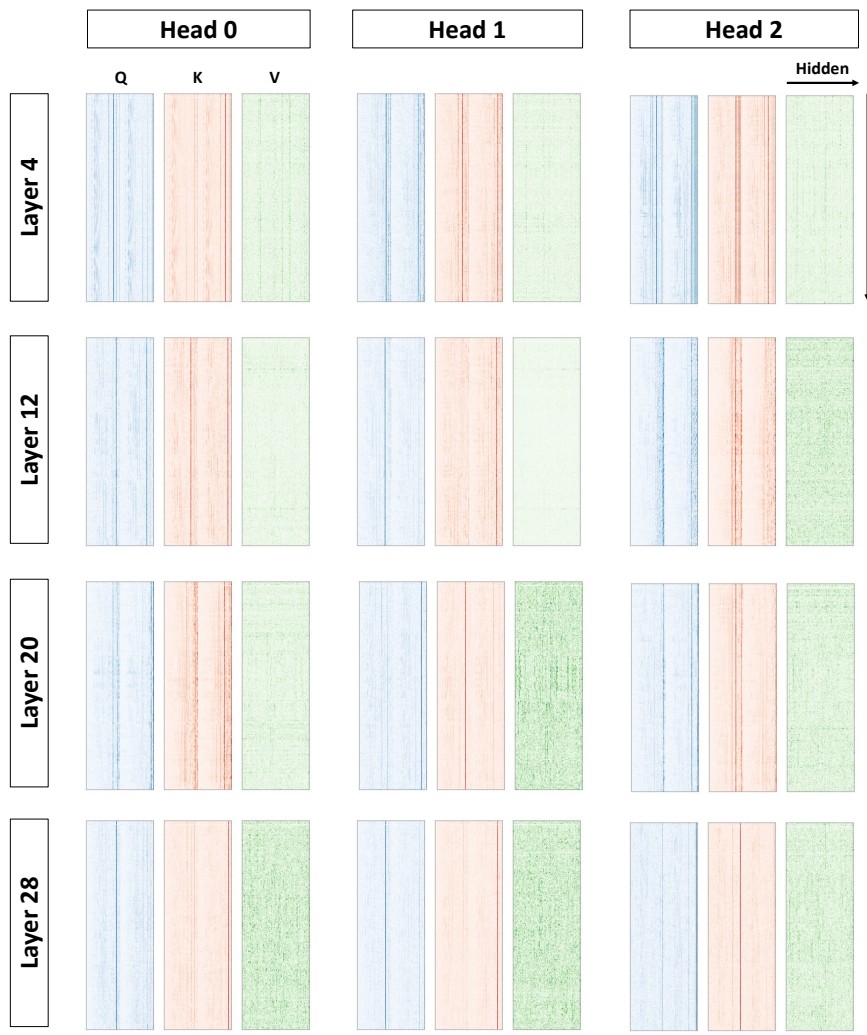

Figure 14: QKV plots for Llama-2-7b-chat.

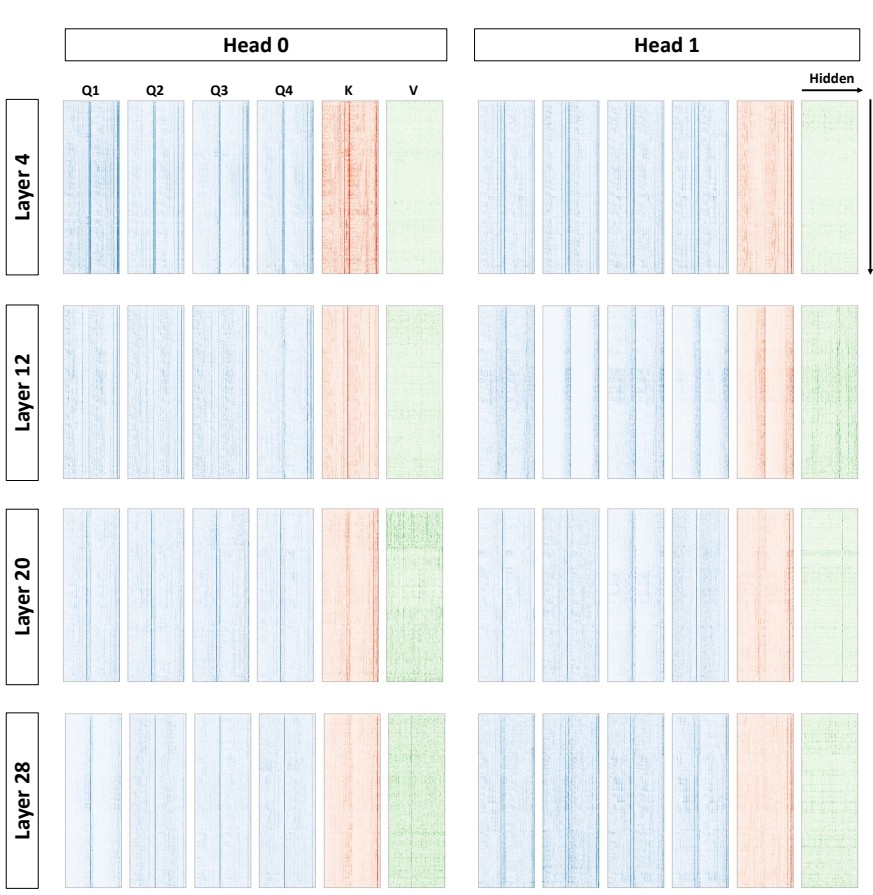

Figure 15: QKV plots for Mistral-7B-Instruct-v0.1.

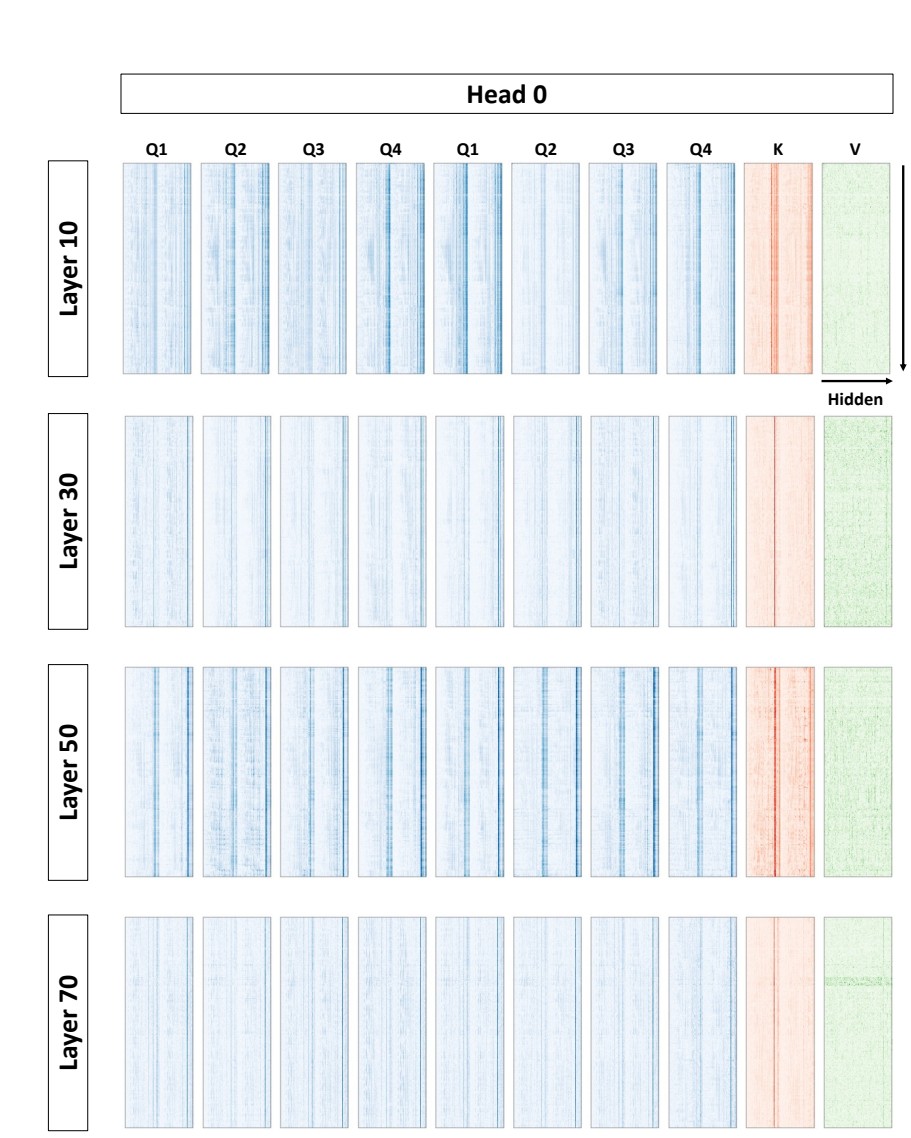

Figure 16: QKV plots for Llama-2-70b-chat.

## C.2 QUANTITATIVE OBSERVATIONS ON OUTLIERS

We conduct quantitative examinations of the outlier severity in the self-attention module. We use the test set of Wikitext-2 (Merity et al., 2017) from Huggingface Datasets (Lhoest et al., 2021) to measure the relative magnitude ratios of the channels in Q, K, and V. Specifically, we take the mean of absolute magnitude along the sequence dimension, and calculate the ratio between the maximum magnitude input-channel(for Query and Key) and the minimum magnitude input-channel(for Query and Key), and aggregate this gain ratio across heads, layers, and input samples:

$$R_Q = \frac{1}{N \cdot L \cdot h} \sum_n \sum_l \sum_h \frac{1}{S} \frac{\max_c \left( \sum_s |Q_{n,l,h,c,s}| \right)}{\min_c \left( \sum_s |Q_{n,l,h,c,s}| \right)}, \tag{3}$$

where $n$ is the sample index, $l$ is the layer index, $h$ is the head index, $c$ is the channel index, and $s$ is the sequence position. We measure this ratio for 4 LLM models and report the results in Table 5. Results show that while Q and K exhibit large outlier ratios, V exhibits relatively low outlier ratios.

Table 5: Query, key, and value outlier magnitude ratios for various models. Query and key exhibit substantial outliers.

| Model | Query | Key | Value |
|-------|-------|-----|-------|
| Llama-2-7b | 23.67 | 29.26 | 3.92 |
| Mistral-7b | 23.82 | 18.12 | 3.93 |
| Llama-2-13b | 32.96 | 40.62 | 4.62 |
| Llama-2-70b | 79.04 | 86.20 | 7.60 |

## D   EXPERIMENTS ON PER-CHANNEL QUANTIZATION OF KEYS

In Section 4.2, we scrutinized and discussed that systematic outlier channels emerge in the keys and queries, which leads to significant quantization errors, degrading the performance. For compatibility with existing off-the-shelf eviction strategies and kernel support, we adopted per-token quantization while mitigating the outlier effect with dynamic outlier awareness. An alternate direction towards mitigating these outliers is *per-channel* quantization, which naturally isolates the outlier channels. Recent works have demonstrated that such a quantization scheme can reduce quantization errors when the direction of quantization and the direction of outlier manifestation align (Heo et al., 2023).

To explore this option, we conduct the experiment in Section 4.2 with per-channel key quantization. However, to impose per-channel dynamic quantization, the caching mechanism must be altered at the implementation level. First, incoming KV pairs must be stored in a temporary buffer until a sufficient amount of KV pairs are accumulated for quantization. Second, additional temporary buffers must be maintained to accumulate important KV pairs and unimportant pairs separately. Third, "evicting" a KV pair from a groupwise per-channel quantized tensor is not straightforward, as the tile size becomes non-uniform. Thus, the underlying eviction policy must be altered. Thus, for compatibility with existing off-the-shelf eviction strategies, we adopted per-token quantization.

Nevertheless, per-channel key quantization is a straightforward approach toward outlier management. To this end, we gauge and analyze the effectiveness of per-channel quantization by conducting experiments with simulated hypothetical per-channel quantization. Our hypothetical quantization scheme quantizes the keys in a per-channel manner with a group size of 64. Since quantization is simulated, we do not reorder or buffer KV pairs and quantize them as-is. Thus, the precision of KV pairs can differ within groups, so that we can maintain the H2O eviction policy. Table 6 shows the line retrieval performance when 20% of the KV pairs are kept in FP16 in the importance cache and 80% of the KV pairs are kept in INTx in the retained cache. The results show that per-channel quantization is effective in preserving the performance, as it isolates outliers. For actual quantization, the underlying eviction policy must be modified to incorporate per-channel quantization, so the performance result may differ. Although the quantization scheme used in this experiment is hypothetical, it demonstrates the possibility of utilizing per-channel quantization to effectively preserve performance if the eviction scheme is modified accordingly, and proper kernel support is provided.

Table 6: Line retrieval accuracy of the retained cache with per-channel key quantization for importance ratio 20%.

| Prec. of Retained KVs | Outlier-Aware | KV Cache Size | Acc. |
|---|---|---|---|
| | ✗ | 36% | 100.0% |
| INT3 | per-token, channel balancer | 38% | 99.8% |
| | per-channel | 38% | 99.4% |
| | ✗ | 32% | 64.0% |
| INT2 | per-token, channel balancer | 33% | 92.6% |
| | per-channel | 33% | 99.2% |

# E    COMPUTATIONAL OVERHEAD OF THE OUTLIER BALANCER

We experimentally assess the computation overhead involved in the utilization of the outlier balancer introduced in Section 4.2. To this end, we measure the end-to-end token generation latency of Llama-2-7b for batch size 32 and input sequence length 512. Detailed experimental settings are provided in Appendix G.7. Table 7 shows that incorporating the balancer results in a minimal increase in token generation latency.

| Balancer | Latency (ms) |
|----------|--------------|
| Yes      | 49.63        |
| No       | 49.60        |

Table 7: Latency comparison with and without the outlier balancer.

# F IMPLEMENTATION DETAILS

## F.1 IMPORTANCE POLICY

As demonstrated in Section 5.3, MiKV can be equipped with various importance policies (Zhang et al., 2023; Li et al., 2024). Here, we provide details on one such policy, the H2O (Zhang et al., 2023) policy, which is equipped throughout our main experiments. This policy decides to maintain important KVs according to the calculated importance of KVs alongside a local window of KVs. First, the importance of KVs is calculated by analyzing the lower triangular attention map produced per self-attention head. The importance is determined by aggregating attention scores across the query dimension.

Let $A \in \mathbb{R}^{N \times N}$ represent the lower triangular attention matrix of a single head in a single layer, where $N$ is the sequence length, and $A_{ij}$ denotes the attention weight from query position $i$ to key position $j$. To compute the importance of each KV, we calculate the accumulated attention score $s_j$ for each key $j$ by summing the attention weights across all queries:

$$s_j = \sum_{i=1}^{N} A_{ij}, \quad j = 1, 2, \ldots, N. \tag{4}$$

This operation aggregates the contributions of key $j$ to all query positions $i$, providing a measure of its overall importance.

## F.2 MIXED PRECISION CACHE MANAGEMENT

We now describe the strategy for managing and accelerating the mixed-precision KV cache. Based on the importance policy, KV pairs are partitioned and re-grouped into high-precision and low-precision groups. This partitioning is feasible due to the following property: after positional encoding is applied, the self-attention mechanism becomes *permutation-invariant* with respect to positions in the KV cache. In other words, as long as KV pairs are permuted together, their order within the cache does not affect the self-attention computation. This enables arbitrary shuffling of KVs for the purpose of grouping them by precision without any functional consequences.

After grouping, each precision group is compressed with a distinct precision level. Within each group, all KVs share the same precision, allowing them to be stored contiguously in memory for efficiency. During the GEMV operation in self-attention, each KV group is accelerated via GPGPU kernels (e.g., CUDA, Triton, etc.) that execute INTn × FP16 GEMV operations. It is important to note that various kernel designs, such as on-the-fly dequantization (Lin et al., 2023; Liu et al., 2024) or lookup-based designs (Park et al., 2024), can be employed to perform the necessary GEMV operations. For our experiments, we adopted an on-the-fly dequantization approach.

# G  DETAILED EXPERIMENTAL SETTINGS

We describe the detailed settings for the experiments conducted in the main paper. We use the Huggingface (Wolf et al., 2019) framework and its generation features for inference. All models are downloaded from the Huggingface Hub and loaded in FP16 format, and all intermediate activations are processed in FP16 unless upcasted by the Huggingface framework (e.g. attention map before softmax). For all experiments, we use deterministic greedy decoding for controlled assessment. All experiments are conducted using internal clusters equipped with Nvidia RTX3090 (24GB VRAM), V100 (32GB VRAM), and A100 (80GB VRAM) GPUs. Experiments on 7b and 13b models were conducted using RTX3090 and V100. Experiments on Llama-2-70b were conducted using 2 A100 GPUs. To measure the accuracy-compression tradeoff, we use the configurations in Table 8.

Table 8: MiKV configurations for main experiments.

| Important KV Prec. | Retained KV Prec. | Importance Ratio | Retained Ratio | Cache Size |
|---|---|---|---|---|
| INT8 | INT4 | 87.5% | 12.5% | 50.0% |
| INT4 | INT3 | 50.0% | 50.0% | 25.0% |
| INT4 | INT2 | 35.0% | 65.0% | 20.0% |
| INT4 | INT2 | 20.0% | 80.0% | 18.1% |
| INT3 | INT2 | 10.0% | 90.0% | 16.3% |

## G.1  GSM8K

We evaluate under a 1-shot chain-of-thought prompt setting, where a full example input is provided in Figure 17. We use the prompt from `https://github.com/FranxYao/chain-of-thought-hub`.

## G.2  HUMANEVAL

We use the 164 evaluation samples provided by Chen et al. (2021). Since we use greedy decoding for evaluation, all samples are generated once each. After generation, we calculate the score using the `evaluate_functional_correctness` command.

## G.3  LINE RETRIEVAL

For the line retrieval task, we use instruction-tuned LLMs to generate expected outputs. Using the code provided by Li et al. (2023a) (`https://github.com/DachengLi1/LongChat`), we synthesize an evaluation set containing 500 samples. A single sample is comprised of an instruction header, 20 lines of index-register context pairs, and a retrieval instruction. The full example input for the experiment is described in Figure 19.

## G.4  MMLU

We evaluate under a 1-shot chain-of-thought prompt setting, where a full example input is provided in in Figure 18. We use the code and prompt in `https://github.com/hendrycks/test`.

## G.5  ALPACAEVAL

For the AlpacaEval (Li et al., 2023b) benchmark, we use the official Github repository of AlpacaEval (`https://github.com/tatsu-lab/alpaca_eval`) and its standard settings. We calculate the win rate by comparing the sequence generated using the compressed cache against the sequence generated with the full cache. We use GPT-4 (OpenAI et al., 2023) (`gpt-4-1106-preview`) as the judge.

## G.6  RULER

For RULER (Hsieh et al., 2024), we use the official Github repository and its default settings (`https://github.com/hsiehjackson/RULER`). We measure all the included minor

sub-tasks and report the average scores within each major task categories. We also report the total weighted average scores.

## G.7 END TO END LATENCY BENCHMARK

To benchmark the end-to-end token generation latency, we use the official Huggingface (Wolf et al., 2019) transformers library to measure the wall-clock latency of Llama-2-7b with a batch size of 32. We observe the latency while varying the sequence length from 512 to 2048, and generate 338 tokens, following the generation length of Kwon et al. (2023). We measure the latency 3 times and report the average value. For H2O, we use the default cuBLAS kernel. For KIVI, we use the CUDA kernel of the official Github repository. For the configuration of MiKV , we utilized INT4 and INT2 with an importance ratio of 50%. All experiments were conducted on NVIDIA A100.

```
Question: There are 15 trees in the grove. Grove workers will plant
trees in the grove today. After they are done, there will be 21 trees.
How many trees did the grove workers plant today?
Let's think step by step
There are 15 trees originally.
Then there were 21 trees after some more were planted.
So there must have been 21 - 15 = 6.
The answer is 6.

Question: Janet's ducks lay 16 eggs per day. She eats three for breakfast
every morning and bakes muffins for her friends every day with four. She
sells the remainder at the farmers' market daily for $2 per fresh duck
egg. How much in dollars does she make every day at the farmers' market?
Let's think step by step
```

Figure 17: Example prompt for GSM8k evaluation.

```
The following are multiple choice questions (with answers) about
abstract algebra.

Find all c in Z_3 such that Z_3[x]/(x^2 + c) is a field.
A. 0
B. 1
C. 2
D. 3
Answer: B

Find the degree for the given field extension Q(sqrt(2), sqrt(3),
sqrt(18)) over Q.
A. 0
B. 4
C. 2
D. 6
Answer:
```

Figure 18: Example prompt for MMLU evaluation.

```
[INST] <<SYS>>
You are a record processing computer. Given a list of records, and a
target <line index>, you retrieve the '<REGISTER_CONTENT>' number.

<</SYS>>

Below is a record of lines I want you to remember. Each line begins with
'line <line index>' and contains a '<REGISTER_CONTENT>' at the end of the
line as a numerical value. For each line index, memorize its
corresponding <REGISTER_CONTENT>. At the end of the record, I will ask
you to retrieve the corresponding <REGISTER_CONTENT> of a certain line
index. Now the record start:

line billowy-schizophrenic: REGISTER_CONTENT is <37977>
line psychotic-cement: REGISTER_CONTENT is <17936>
line daffy-pancake: REGISTER_CONTENT is <31235>
line exclusive-bough: REGISTER_CONTENT is <28484>
line enthusiastic-navigation: REGISTER_CONTENT is <12927>
line handsome-variability: REGISTER_CONTENT is <35756>
line enchanting-thrust: REGISTER_CONTENT is <12197>
line sour-hippopotamus: REGISTER_CONTENT is <16604>
line faithful-tabernacle: REGISTER_CONTENT is <20711>
line picayune-cookie: REGISTER_CONTENT is <20822>
line wee-basics: REGISTER_CONTENT is <41007>
line forgetful-struggle: REGISTER_CONTENT is <45999>
line cagey-cargo: REGISTER_CONTENT is <8069>
line childlike-polyp: REGISTER_CONTENT is <27732>
line inconclusive-flesh: REGISTER_CONTENT is <39135>
line delightful-location: REGISTER_CONTENT is <12214>
line courageous-viability: REGISTER_CONTENT is <23079>
line scandalous-laboratory: REGISTER_CONTENT is <2510>
line mere-affect: REGISTER_CONTENT is <34561>
line annoyed-armrest: REGISTER_CONTENT is <27869>

Now the record is over. Tell me what is the <REGISTER_CONTENT> in line
inconclusive-flesh? I need the number. [/INST]
```

Figure 19: Example prompt for the line retrieval task.

# H  MEMORY FOOTPRINT ANALYSIS

We report the reduction in KV cache memory footprint for the models used in our experiments. We assess the memory consumption for batch size 8 and sequence length 4096. Table 9 indicates that MiKV significantly reduces memory usage for models of varying sizes and GQA availability.

Table 9: Memory footprint comparison between the full KV cache and MiKV. We compare the reduction on models of varying sizes and GQA availability for batch size 8 and sequence length 4K.

| Model | GQA | Cache Size | Memory | MMLU |
|---|---|---|---|---|
| Llama-2-7b | ✗ | 100% | 34.36GB | 44.0% |
|  |  | 25% | 8.59GB | 43.9% |
|  |  | 20% | 6.87GB | 42.7% |
| Mistral-7b | ✓ | 100% | 8.59GB | 61.0% |
|  |  | 25% | 2.15GB | 60.9% |
|  |  | 20% | 1.72GB | 60.7% |
| Llama-2-13b | ✗ | 100% | 53.69GB | 52.7% |
|  |  | 25% | 13.42GB | 52.9% |
|  |  | 20% | 10.74GB | 52.6% |
| Llama-2-70b | ✓ | 100% | 17.18GB | 67.7% |
|  |  | 25% | 4.30GB | 67.8% |
|  |  | 20% | 3.44GB | 67.8% |

# I AlpacaEval Results

We further evaluate the generation quality of MiKV on a chatbot benchmark for instruction-tuned models by measuring AlpacaEval (Li et al., 2023b) win rate against a full cache, FP16-weight model for Llama-2-70b-chat. Results in Table 10 show that MiKV does not exhibit a drop in win rate, for cache sizes as small as 25%. Moreover, MiKV exhibits minimal degradations when combined with 4-bit weight-only quantization (Lee et al., 2023).

Table 10: AlpacaEval win rate for Llama-2-70b-chat combined with MiKV and FlexRound (Lee et al., 2023).

| W | A | KV | Cache size | Win rate |
|------|------|------|------------|----------|
| FP16 | FP16 | Full | 100% | 50.0% |
| FP16 | FP16 | MiKV | 25%
20% | 51.1%
48.6% |
| INT4 | FP16 | MiKV | 25%
20% | 49.9%
46.5% |

## J LIMITATIONS

The acceleration provided by our work on KV cache compression is limited to the speedup of the generation phase, where self-attention is memory-bound. The scope of our work does not include the acceleration for the compute-bound prefill phase and is orthogonal to works such as Dao et al. (2022).

Due to limited computation resources, our benchmark experiments are conducted with deterministic greedy decoding. The performance of our proposed method and other baselines are currently not tested under random sampling decoding or beam search decoding.

## K BROADER IMPACT

This paper presents a work in LLM KV cache compression for efficient and accelerated inference by mitigating the memory footprint of the KV cache. We examine existing KV cache compression methods in the context of contextual safety, and observe that cache eviction can result in undesired model behavior (system prompt breach, response incoherence, hallucinations, and context detail loss), inducing social impacts. We propose our method to mitigate the potential contextual safety issues caused by KV cache compression while preserving model performance. We project that our work can provide insights contextual safety in compressing the KV cache of transformer-based LLMs for efficient deployment. On the other hand, a possible negative impact of our work may be that an LLM user may decide to tolerate a decline in reliability and generation quality for the sake of efficiency.

# L  MAIN BENCHMARK RESULTS OF LARGER MODELS

We further evaluate the performance of MiKV on GSM8K, HumanEval, Line Retrieval, and MMLU for `Llama-2-13b`, a larger LLM. Experimental results in Figure 20 show that MiKV achieves better accuracy-compression tradeoff compared to baselines also for larger models.

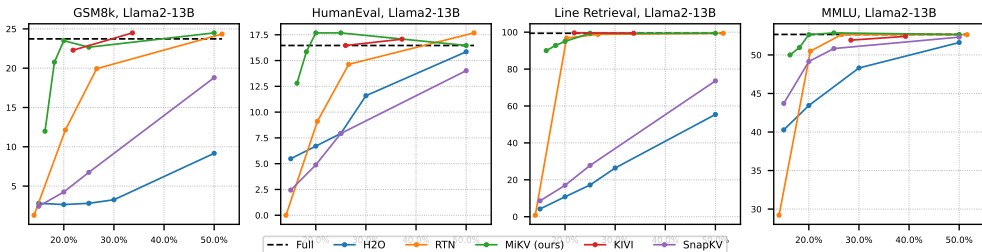

Figure 20: Performance results of MiKV compared to baselines on GSM8k, HumanEval, Line Retrieval, and MMLU for Llama-2-13b. The $x$ axes represent the compressed KV cache size (%). The $y$ axes represent the benchmark accuracy (%). We compare our method (MiKV) with eviction (H2O, SnapKV) and quantization (RTN, KIVI).

# M    ADDITIONAL RULER RESULTS

We conduct additional experiments on the RULER (Hsieh et al., 2024) benchmark with Longchat-7b on 8k context length. As shown in Table 11, MiKV successfully maintains the benchmark performance, while achieving further compression.

Table 11: RULER benchmark results for 8K context length.

| Method | Cache size | Synthetic | | | | Real | wAvg. |
| | | Single retrieval | Multi retrieval | Multi-hop | Aggregation | QA | |
|---|---|---|---|---|---|---|---|
| Full | 100% | 99.73 | 85.50 | 89.08 | 48.70 | 52.20 | 78.28 |
| KIVI-4 | 28% | 99.00 | 85.51 | 89.52 | 49.49 | 52.50 | 78.31 |
| KIVI-2 | 17% | 51.73 | 35.59 | 64.20 | 58.04 | 50.10 | 47.20 |
| H2O | 25% | 13.53 | 12.74 | 34.56 | 50.64 | 47.00 | 25.70 |
| MiKV | 25% | 99.00 | 85.37 | 89.56 | 50.28 | 52.20 | 78.34 |

# N    LONGBENCH RESULTS

We use the official GitHub repository of LongBench (Bai et al., 2024) and Llama-2-7b to test the integrity of our proposed method on non-synthetic long context inputs. Experimental results in Table 12 demonstrate that our method is capable of compressing the KV cache while preserving accuracy even under long-context conditions.

Table 12: LongBench benchmark results.

| Method | Cache size | Tasks | | | | | | | | Average |
| | | Trivia QA | Qasper | TREC | Samsum | LCC | RepoBench-P | QMSum | Multi-News | |
|---|---|---|---|---|---|---|---|---|---|---|
| FULL | 100% | 83.26 | 22.05 | 64.0 | 41.31 | 58.24 | 52.28 | 20.76 | 26.23 | 46.02 |
| KIVI-4 | 28% | 84.78 | 20.44 | 58.50 | 38.78 | 52.26 | 49.74 | 19.67 | 25.94 | 43.76 |
| KIVI-2 | 17% | 83.99 | 19.45 | 58.50 | 38.20 | 51.00 | 47.53 | 19.35 | 25.04 | 42.88 |
| MiKV | 25% | 83.34 | 20.72 | 64.00 | 43.93 | 57.32 | 52.51 | 20.51 | 26.09 | 46.05 |
| MiKV | 20% | 83.51 | 20.23 | 64.00 | 43.00 | 58.16 | 52.06 | 20.44 | 25.49 | 45.86 |

## O    THROUGHPUT ANALYSIS

Here, we conduct additional experiments comparing the throughput of MiKV with baselines using the llama-2-7b model. Fixing the sequence length to 1024, we measure the throughput (tokens/s) of the model across varying batch sizes of 16, 32, and 64. The results are demonstrated in Figure 21. Experimental results demonstrate that MiKV achieves increased throughput compared to the FP16 full-cache, with the improvement becoming more pronounced as the batch size increases.

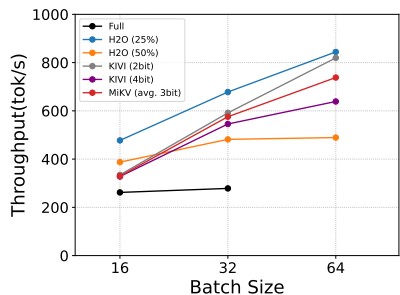

Figure 21: End-to-end generation throughput (tokens/s) of MiKV compared to baselines.

Unlike KIVI, which uses a uniform bitwidth quantization, MiKV employs a mixed precision strategy with an average precision of 3 bits, assigning 4 bits to important KVs and 2 bits to preserved KVs. This approach balances accuracy and compression, resulting in practical speed improvements.

In contrast, H2O, which adopts an eviction-based approach, achieves the highest throughput by omitting computation for evicted KVs. However, this comes at the cost of significant accuracy degradation at similar compression ratios (as demonstrated in Figure 6), limiting its applicability. Even when H2O uses a conservative compression ratio (50%) to preserve accuracy, it still lags behind MiKV in both accuracy and throughput. These results highlight MiKV as a more effective solution, maintaining accuracy while optimizing throughput and memory.

