# OpenReview forum: "Don’t Discard, but Keep It Small: Context-Preserving KV Cache Compression with Importance-Aware Adaptive Precision"
_ICLR.cc/2025/Conference — Submitted to ICLR 2025_

### Official Review · Reviewer_j7KD · 2024-10-19

**Soundness:** 3
**Presentation:** 2
**Contribution:** 3
**Rating:** 6
**Confidence:** 4

**Summary:**

This paper addresses the issue that token eviction-based KV cache compression strategies can lead to the loss of critical context details, resulting in unintended generation outputs. The authors propose a mixed-precision KV cache (MiKV), which compresses KV pairs using a mixed precision approach rather than discarding them. The method preserves more important KV pairs in higher precision and stores others in lower precision. Experiments show that MiKV outperforms token eviction methods (e.g., H2O, SnapKV) and quantization approaches (e.g., KIVI) on selected benchmarks.

**Strengths:**

1. KV cache compression is a crucial research topic for large language models.
2. The idea of MiKV is simple and easy to implement.
3. Experimental results indicate that MiKV effectively retains critical details compared to eviction-based methods.

**Weaknesses:**

1. The novelty is somewhat limited. While the method is effective, the concept of identifying important tokens in the KV cache is not new, and mixed-precision quantization is a widely used technique in LLM quantization.
2. The paper would be clearer if it provided more detail on how important tokens are selected, rather than simply referencing prior research. This would make the paper more self-contained.

**Questions:**

1. I found Figure 1 a bit confusing. Does the quantization scheme for a token change as generation progresses? For instance, can a token in the cache shift from INT4 to INT2 during later stages of generation?
2. In Section 5, MiKV is compared with another KV cache quantization approach, KIVI. KIVI uses per-channel quantization for keys and per-token quantization for values, as outlined in the original paper. Was this setting preserved in your experiments? If not, the comparison might not be fair.

---

> ### Author Response · Authors · 2024-11-22
> **Response to Reviewer j7KD (1)**
>
> ---
>
> We sincerely appreciate your feedback in improving our manuscript. We address the reviewer’s concerns below.
>
> ---
>
> > **[W1.] The novelty is somewhat limited. While the method is effective, the concept of identifying important tokens in the KV cache is not new, and mixed-precision quantization is a widely used technique in LLM quantization.**
>
> Thank you for your constructive feedback. The novelty of our work lies in two key contributions:
>
> 1. We identify and analyze previously **unrecognized hazards** exhibited by KV eviction methods, which were widely considered safe.
> 2. We introduce a novel **mixed-precision approach** for KV compression to address these risks by preserving the evicted KVs in low precision and compressing important KVs in high precision.
>
> First, as the reviewer has stated, the concept of identifying important tokens in the KV cache has been previously explored by works on KV eviction[1, 2]. However, they report seemingly minimal accuracy degradation even when maintaining less than 20% of important KVs (by evicting 80% of KVs), compressing the cache size down to 20% of the original (e.g. Figure 2(d) of [1]). However, contrary to the prevailing reports that KV eviction is safe for such compression regimes, our analyses and experiments reveal that KV eviction can lead to unexpected safety risks and context damage, even under compression regimes like 50% that were previously considered reliable (Figures 2 and 3 of the manuscript). In particular, our analysis of intrinsic issues of sparsity (Section 3.2) highlights the inherent difficulty of predicting which KVs will remain important in the future, demonstrating the unreliability of eviction. To the best of our knowledge, **we are the first to reveal and analyze such hidden risks in KV cache eviction**, especially from the perspective of context damage.  By doing so, we aim to highlight scenarios where such risks (safety breaches, hallucinations, etc.) render these methods impractical from the perspective of real-world LLM services. Through evaluations across multiple benchmarks, we demonstrate how KV eviction can critically impact the reliability of LLMs in practice. Our findings emphasize the need for a more thorough evaluation of KV cache eviction methods across diverse aspects, as their risks may outweigh the benefits.
>
> Second, motivated by these findings, we propose a novel KV compression approach to mitigate the risks of KV eviction by employing mixed-precision. While the concept of mixed-precision quantization has been applied to LLMs before (e.g. [3]) to control outliers in the model, to the best of our knowledge, **we are the first to apply mixed-precision** quantization to LLM **KV cache compression**. Unlike eviction methods, our method (MiKV) recognizes that while “important” KVs are critical for LLM performance, preserving only these KVs is insufficient to ensure contextual safety. The information encoded in "less important" KVs also plays a vital role in maintaining context and must be efficiently preserved.
>
> On the other hand, other quantization approaches that use uniform precision do not take “important” KVs into consideration, resulting in significant accuracy loss under aggressive low-precision or inefficient compression rates under conservative high-precision. However, MiKV selectively compresses KVs using mixed-precision quantization based on their importance, ensuring that critical information is retained while minimizing memory usage.
>
> Our method enables an effective balance between cache compression and contextual integrity, addressing a trade-off that previous methods have overlooked.
>
> ---
>
> > **[W2.] The paper would be clearer if it provided more detail on how important tokens are selected, rather than simply referencing prior research. This would make the paper more self-contained.**
>
> Thank you for your suggestion. We have revised our manuscript to provide the details on how important tokens are selected in **Appendix F**.
>
> ---

---

> ### Author Response · Authors · 2024-11-22
> **Response to Reviewer j7KD (2)**
>
> ---
>
> > **[Q1.] I found Figure 1 a bit confusing. Does the quantization scheme for a token change as generation progresses? For instance, can a token in the cache shift from INT4 to INT2 during later stages of generation?**
>
> The behavior of the quantization scheme depends on the design choice. After the initial quantization based on importance, the precision of a KV pair can either remain fixed or be adjusted dynamically (re-quantized). In our main experiments, we adopted the latter approach, where high-precision KV pairs could shift to lower precision through re-quantization. Although re-quantization noise is introduced, our experiments demonstrate that MiKV effectively preserves accuracy while compressing the KV cache.
>
> ---
>
> > **[Q2.] In Section 5, MiKV is compared with another KV cache quantization approach, KIVI. KIVI uses per-channel quantization for keys and per-token quantization for values, as outlined in the original paper. Was this setting preserved in your experiments? If not, the comparison might not be fair.**
>
> As mentioned by the reviewer, KIVI[4] employs per-channel quantization for keys. Therefore, in our experiments with KIVI, we ensured that per-channel key quantization was used as specified.
>
> ---
>
> Once again, we sincerely appreciate your time and effort in reviewing our paper. If you have any remaining issues or concerns, please do not hesitate to bring them to our attention.
>
> ---
>
> **References:**
>
> [1] Zhang et al. “H2O: Heavy-Hitter Oracle for Efficient Generative Inference of Large Language Models”. NeurIPS 2023.
>
> [2] Li et al. “SnapKV: LLM Knows What You are Looking for Before Generation". NeurIPS 2024.
>
> [3] Dettmers et al. "Gpt3. int8 (): 8-bit matrix multiplication for transformers at scale." NeurIPS 2022.
>
> [4] Liu et al. “KIVI: A Tuning-Free Asymmetric 2bit Quantization for KV Cache”. ICML 2024.
>
> ---

---

> > ### Comment · Reviewer_j7KD · 2024-11-25
> >
> > Thank you for the response and clarification. I believe my original assessment remains fair and would like to keep my score.

---

> > > ### Author Response · Authors · 2024-12-02
> > >
> > > We deeply appreciate your time to review our paper and response. Thank you for your support and thoughtful feedback.

---

### Official Review · Reviewer_14Mp · 2024-10-28

**Soundness:** 3
**Presentation:** 3
**Contribution:** 2
**Rating:** 5
**Confidence:** 3

**Summary:**

The paper proposes the **MiKV** (Mixed-precision KV cache) strategy, an adaptive compression method for KV caching in LLMs to optimize memory usage without compromising performance. Unlike traditional eviction methods that discard less important KV pairs, potentially degrading context retention, **MiKV** preserves evicted pairs at lower precision and keeps crucial pairs at high precision, balancing memory efficiency with context integrity. The study identifies that cache eviction often leads to context loss, risking issues like safety breaches, hallucinations, and incoherent outputs. By introducing an outlier-aware quantization and importance-based precision control, **MiKV** maintains the generation quality across multiple benchmarks, achieving higher compression ratios and comparable performance to full-cache models. Experimental results indicate **MiKV**’s robustness in handling long contexts and minimizing memory footprint on GPUs.

**Strengths:**

1. **MiKV** effectively mitigates the risk of context loss by preserving all KV pairs in varying precisions, which prevents common issues like hallucinations and prompt breaches seen in eviction-based methods. This approach maintains generation quality even with high compression ratios.

2. The **MiKV** experiments are relatively thorough, covering diverse benchmarks like GSM8k, HumanEval, Line Retrieval, and MMLU to demonstrate effectiveness across tasks.

3. The plug-and-play design of **MiKV** enhances its applicability, making it a versatile tool for LLM deployment in memory-constrained environments and bringing reasonable performance improvements

**Weaknesses:**

1. The paper mentioned that **MiKV** can be used off-the-shelf integrated with other important token selection policies such as H2O but didn't clearly state what eviction policies were used in experiments, especially Table 1, Table 3, and Figure 6. The experiments on larger models are limited.

2. The improvement of **MiKV** on SnapKV is quite limited, showing that the mixed-quantization approach may not be universally effective on different important-token-selection policies, which shows that the most crucial component is the choice of selection policy of important tokens but not the mixed-precision strategy.

3. In the latency analysis, the paper didn't measure the end-to-end latency of quantization method RTN, which achieves comparable performance in several settings in Figure 6.

**Questions:**

1. How does the **MiKV** perform GSM8k, HumanEval, Line Retrieval, and MMLU in tasks on larger models? What's the change of generation latency costed by larger models applied with **MiKV** compared to other approaches?

2. What's the implication behind the limited improvement of **MiKV** applied on SnapKV?

---

> ### Author Response · Authors · 2024-11-22
> **Response to Reviewer 14Mp (1) [UPDATED]**
>
> ---
>
> We sincerely appreciate your feedback in improving our manuscript. We address the reviewer’s concerns below.
>
> ---
>
> > **[W1, Q1] The paper mentioned that MiKV can be used off-the-shelf integrated with other important token selection policies such as H2O but didn't clearly state what eviction policies were used in experiments, especially Table 1, Table 3, and Figure 6.**
>
> Thank you for your insightful comment. For experiments in the main manuscript (including Table 1, Table 3, and Figure 6), we used the cumulative attention score as the importance policy (H2O[1]), which was stated in Line 409 of the main manuscript. For Table 4 of the manuscript, we conduct an ablation study on the importance policy by measuring the GSM8K performance of MiKV when using different policies. Thus, for this experiment, 2 different policies[1, 2] were used.
>
> ---
>
> > **[W1, Q1] The experiments on larger models are limited. How does the MiKV perform GSM8k, HumanEval, Line Retrieval, and MMLU in tasks on larger models? What's the change of generation latency costed by larger models applied with MiKV compared to other approaches?**
>
> **[UPDATED]** Thank you for your feedback in improving our manuscript. Due to constraints on computational resources, we mainly conducted our experiments using 8B-scale models. However, to address the reviewer’s concerns, we extend our accuracy and latency evaluations to **Llama-2-13b**, a **larger model**. We conducted additional experiments on **GSM8K, HumanEval, Line Retrieval, MMLU**, and updated our manuscript to include these experiments in **Appendix L**. Among the four benchmarks, we report the GSM8K accuracy for MiKV and baselines below (please kindly refer to Appendix L for full results for all benchmarks):
>
> | Method | Cache Size | GSM8K |
> | --- | --- | --- |
> | Full | 100% | 23.73% |
> | H2O | 20% | 2.65% |
> | RTN | 20% | 12.13% |
> | KIVI | 22% | 22.29% |
> | SnapKV | 20% | 4.25% |
> | MiKV | 20% | 23.50% |
>
> Experimental results show that also for larger models, MiKV achieves better accuracy-compression tradeoff compared to baselines.
>
> Also, to address the reviewer’s concerns on the **latency** of **larger models**, we extend our latency evaluations to include comparisons between llama-2-7b and llama-2-13b. For batch size 32 and sequence length 1024, we vary the model size from 7b to 13b and measure the latency of MiKV and other baselines:
>
> | Method | Model Size |  |
> | --- | --- | --- |
> |  | **7b** | **13b** |
> | Full Cache | 114.9 (ms) | OOM |
> | H2O (50%) | 66.4 | 116.7 |
> | H2O (25%) | 47.2 | 73.9 |
> | KIVI (2bit) | 54.1 | 77.3 |
> | KIVI (4bit) | 58.7 | 93.2 |
> | RTN (2bit) | 50.9 | 79.2 |
> | RTN (4bit) | 62.1 | 100.8 |
> | MiKV (avg. 3bit) | 55.6 | 84.5 |
>
> Experimental results show that the latency of MiKV (avg. 3bit) falls between that of KIVI (2bit) and KIVI (4bit), confirming that the mixed-precision quantization of MiKV is successfully accelerated across different model sizes while achieving a meaningful balance between accuracy and latency. Importantly, when scaling up from llama-2-7b to llama-2-13b, we observed that the trends in latency and accuracy remained **consistent**. This indicates that MiKV's approach to mixed-precision provides a robust accuracy-latency trade-off, even as model size increases.
>
> These results show the applicability of MiKV across diverse model sizes and tasks, reinforcing its effectiveness in balancing performance and efficiency in LLM deployments.
>
> ---

---

> > ### Comment · Reviewer_14Mp · 2024-11-25
> >
> > Thanks for your responses. I would like to keep both of my scores and confidence given the incomplete list of experiments.

---

> > > ### Author Response · Authors · 2024-12-02
> > > **Gentle Reminder**
> > >
> > > Dear Reviewer **14Mp**,
> > >
> > > We greatly appreciate the time and effort you have dedicated to reviewing our paper.
> > > With the discussion period ending in less than **24 hours**, we kindly ask you to read our responses, as we have **fully completed** your requested experiments.
> > >
> > > To address your concerns faithfully, we conducted all the experiments you suggested. Specifically, we performed evaluations on a larger model (**Llama-2-13B**) across all main benchmarks (GSM8K, HumanEval, Line Retrieval, and MMLU), with **full results provided in Appendix L**. Furthermore, we conducted **latency benchmark** on Llama-2-13B as well (please see our response above).
> > >
> > > In addition, during the rebuttal period, we also conducted throughput evaluations (**Appendix O**), which further substantiate the effectiveness of our methodology. Furthermore, we conducted comparisons with additional baseline methods[3, 4] to further validate the effectiveness of our approach. The results are as follows:
> > >
> > > | Method | Cache Size | GSM8K | Line Retrieval |
> > > | --- | --- | --- | --- |
> > > | **Full** | 100% | 35.2% | 100.0% |
> > > | **kNorm [3]** | 50% | 1.4% | 0.2% |
> > > |  | 25% | 1.0% | 0.0% |
> > > |  | 20% | 0.7% | 0.0% |
> > > | **TOVA [4]** | 50% | 29.3% | 64.0% |
> > > |  | 25% | 12.5% | 10.4% |
> > > |  | 20% | 8.1% | 5.4% |
> > > | **MiKV (ours)** | 50% | 35.5% | 100.0% |
> > > |  | 25% | 36.0% | 100.0% |
> > > |  | 20% | 33.4% | 97.8% |
> > >
> > > Experimental results demonstrate that additional eviction baselines also suffer from large accuracy degradation, whereas MiKV achieves a markedly better balance between accuracy and compression. These experimental results further confirm the shared challenges faced by eviction methods and highlight the robustness of our approach.
> > >
> > > If you have any remaining questions or concerns, please feel free to reach out to us.
> > >
> > > Sincerely,
> > >
> > > Authors
> > >
> > > ---
> > >
> > > **References**
> > >
> > > [3] Devoto et al. “A Simple and Effective L2 Norm-Based Strategy for KV Cache Compression". EMNLP 2024.
> > >
> > > [4] Oren et al. “Transformers are Multi-State RNNs”. EMNLP 2024.
> > >
> > > ---

---

> ### Author Response · Authors · 2024-11-25
> **Response to Reviewer 14Mp (2) [UPDATED]**
>
> ---
>
> > **[W2, Q2.] The improvement of MiKV on SnapKV is quite limited, showing that the mixed-quantization approach may not be universally effective on different important-token-selection policies, which shows that the most crucial component is the choice of selection policy of important tokens but not the mixed-precision strategy.**
>
> There could be a misunderstanding regarding the key component driving the performance of MiKV. The most critical factor is the **use of mixed precision**, not the choice of the importance-token selection policy. This is evident when comparing MiKV equipped with SnapKV policy (mixed precision) to SnapKV itself (not mixed precision).
>
> | Method | Importance Policy | Mixed Precision | Cache Size | GSM8K |
> | --- | --- | --- | --- | --- |
> | H2O | H2O | No | 20% | 2.35% |
> | MiKV | H2O | Yes | 20% | 33.81% |
> | SnapKV | SnapKV | No | 20% | 6.97% |
> | MiKV | SnapKV | Yes | 20% | 33.43% |
>
> As shown in the table above (the results are also displayed in Table 4 of the manuscript), SnapKV's accuracy drops sharply, whereas MiKV equipped with SnapKV maintains high accuracy (31.46% difference). A similar trend can be observed when comparing H2O to MiKV equipped with H2O's policy (26.46% difference). When mixed precision is consistently applied while varying the important token selection policies, performance does not degrade significantly (0.38% difference). Thus, the most crucial component is the mixed precision strategy, not the specific choice of the importance-token selection policy.
>
> ---
>
> > **[W3.] In the latency analysis, the paper didn't measure the end-to-end latency of quantization method RTN, which achieves comparable performance in several settings in Figure 6.**
>
> **[UPDATED]** We sincerely thank the reviewer for the feedback, which has helped us improve our manuscript. To address the reviewer’s concern, we conducted latency measurements for **RTN** and **included** these results in **Figure 7** of the revised manuscript. ~~Currently,~~ experiments are completed and included for all sequence lengths  ~~1024 and 2048 (we are waiting for resources to finish sequence length 512, and will update the results as soon as possible)~~.
>
> Results indicate that RTN achieves either slightly faster or comparable latency compared to KIVI at the same precision levels. Since they both use uniform precision, the relative latency between the two methods may vary from optimization.
>
> ---
>
> Once again, we sincerely appreciate your time and effort in reviewing our paper. If you have any remaining issues or concerns, please do not hesitate to bring them to our attention.
>
> ---
>
> **References**
>
> [1] Zhang et al. “H2O: Heavy-Hitter Oracle for Efficient Generative Inference of Large Language Models”. NeurIPS 2023.
>
> [2] Li et al. “SnapKV: LLM Knows What You are Looking for Before Generation". NeurIPS 2024.
>
> ---

---

> ### Author Response · Authors · 2024-11-25
> **Update on additional experiments**
>
> **Dear Reviewer 14Mp,**
>
> We sincerely apologize for the delay in conducting the remaining experiments to address your concerns. We understand that your experimental inquiries regarding [**W1], [Q1], and [W3]** are valuable and require timely attention. However, due to resource constraints, it took us additional time to conduct the necessary experiments.
>
> We have now completed the accuracy and latency experiments on **larger model** mentioned in **[W1, Q1]** and have updated the corresponding part in our author response accordingly. Also, we have completed the latency experiments for **RTN** (experiments completed and included for sequence length 1024 and 2048; waiting for resource allocation to complete 512) and updated **[W3] and Figure 7** of our manuscript to include these results. In summary, in the author response comments above, **we have updated [W1, Q1], and [W3].**
>
> We are truly grateful for your insightful feedback in helping us improve our work. We hope this updated response with new experimental data resolves your concerns and provides clarity on the raised questions.
>
> Sincerely,
>
> Authors

---

> ### Author Response · Authors · 2024-11-26
> **Further update on additional experiments**
>
> **Dear Reviewer 14Mp,**
>
> We finished the last remaining experiment (latency for RTN for sequence length 512 for llama-2-7b) and updated **[W3] and Figure 7** of our manuscript to include these results. We hope this updated response with the complete list of requested experimental data resolves the reviewer’s concerns.
>
> Sincerely,
>
> Authors

---

### Official Review · Reviewer_Bs5E · 2024-11-01

**Soundness:** 2
**Presentation:** 2
**Contribution:** 1
**Rating:** 3
**Confidence:** 3

**Summary:**

The paper proposes a new KV cache compression method, called MiKV. MiKV quantizes unimportant KV cache into lower bit presentation while maintaining more important KV cache in higher bit presentation. The paper claims superior performance against existing baselines.

**Strengths:**

- The proposed channel balancer to avoid per-channel outliers is effective.

**Weaknesses:**

1. The technical novelty of the paper is limited. In my opinion, there is little to no technical novelty presented in the paper. The proposed method is simply a combination of KV cache dropping method + quantization, which is in plain sight to notice given the orthogonality of KV cache dropping and quantization.
Moreover, the proposed combination does not offer any substantial gain in term of performance, compared with quantization-only baselines in the experiments section, e.g. KIVI.
2. The empirical study on the detrimental effect of kv cache dropping offers little insights. Most of the claims are easy to notice given the nature of kv cache dropping methods.
Moreover, the claims in section 3 are not supported by any experiment results.
3. KV cache dropping baselines are not enough. For example, H20 is considered to be old now given the fast pace development in this field, as well as a weak baseline [1]. There are more recent KV cache dropping baselines that are strong on long-context tasks [2] [3].

The proposed method is still in early stage and requires a major improvement. Thus, I believe the submission is not ready for publication and recommend rejection.

[1] KV Cache Compression, But What Must We Give in Return? A Comprehensive Benchmark of Long Context Capable Approaches. Yuan et al., https://arxiv.org/pdf/2407.01527
[1] MInference 1.0: Accelerating Pre-filling for Long-Context LLMs via Dynamic Sparse Attention. Jiang et al., https://arxiv.org/pdf/2407.02490
[2] Razorattention: Efficient kv cache compression through retrieval heads, 2024. Tang et al., https: //arxiv.org/abs/2407.15891

**Questions:**

Please see weaknesses.

---

> ### Author Response · Authors · 2024-11-22
> **Response to Reviewer Bs5E (1)**
>
> ---
>
> We sincerely appreciate your feedback in improving our manuscript. We address the reviewer’s concerns below.
>
> ---
>
> > **[W1.] The technical novelty of the paper is limited. In my opinion, there is little to no technical novelty presented in the paper. The proposed method is simply a combination of KV cache dropping method + quantization, which is in plain sight to notice given the orthogonality of KV cache dropping and quantization. Moreover, the proposed combination does not offer any substantial gain in terms of performance, compared with quantization-only baselines in the experiments section, e.g. KIVI.**
>
> Thank you for your constructive feedback. The novelty of our work lies in two key contributions:
>
> 1. We identify and analyze previously **unrecognized hazards** exhibited by KV eviction methods, which were widely considered safe.
> 2. We introduce a **novel mixed-precision approach** to address these risks by preserving the evicted KVs in low precision and compressing important KVs in high precision.
>
> First, previous works on KV cache eviction [4, 5] report seemingly minimal accuracy degradation even when evicting more than 80% of KVs, compressing the cache size down to **20%** of the original (e.g. Figure 2(d) of [4]). However, contrary to the prevailing reports that KV eviction is safe for such compression regimes, our analyses and experiments reveal that KV eviction can lead to unexpected safety risks and context damage, even under compression regimes like **50%** that were previously considered reliable (Figures 2 and 3 in the main paper). In particular, our analysis of intrinsic issues of sparsity (Section 3.2) highlights the inherent difficulty of predicting which KVs will remain important in the future, demonstrating the unreliability of eviction. To the best of our knowledge, **we are the first to reveal and analyze such hidden risks in KV cache eviction**, especially from the perspective of context damage.  By doing so, we aim to highlight scenarios where such risks (safety breaches, hallucinations, etc.) render these methods impractical from the perspective of real-world LLM services. Through evaluations across multiple benchmarks, we demonstrate how KV eviction can critically impact the reliability of LLMs in practice. Our findings emphasize the need for a more thorough evaluation of KV cache eviction methods across diverse aspects, as their risks may outweigh the benefits.
>
> Second, motivated by these findings, we propose a novel KV compression approach to mitigate the risks of KV eviction by employing mixed-precision. To the best of our knowledge, **we are the first to apply mixed-precision** quantization to LLM KV cache compression. Unlike eviction methods, our method (MiKV) recognizes that while “important” KVs are critical for LLM performance, preserving only these KVs is insufficient to ensure contextual safety. The information encoded in "less important" KVs also plays a vital role in maintaining context and must be efficiently preserved.
>
> On the other hand, other quantization approaches such as KIVI[6] which use **uniform precision** do not take “important” KVs into consideration, resulting in **accuracy loss** under aggressive low-precision or **inefficient compression** rates under conservative high-precision, as demonstrated in **Figure 6** and **Table 3**. However, MiKV selectively compresses KVs using mixed-precision quantization based on their importance, ensuring that critical information is retained while minimizing memory usage. Thus, MiKV achieves a **favorable accuracy-compression trade-off** compared to both eviction methods and quantization methods.
>
> ---

---

> ### Author Response · Authors · 2024-11-22
> **Response to Reviewer Bs5E (2)**
>
> ---
>
> > **[W2.] The empirical study on the detrimental effect of kv cache dropping offers little insights. Most of the claims are easy to notice given the nature of kv cache dropping methods. Moreover, the claims in section 3 are not supported by any experiment results.**
>
> Thank you for your comment. The reviewer’s point that accuracy degradation due to KV cache eviction may be “easy to notice” could be valid in certain extreme cases (for instance, when 99.9% of KVs are dropped, significant performance loss is expected). However, previous studies such as [4] have suggested that KV cache eviction is stable even for aggressive compression regimes, that more than 80% of the KV cache can be evicted with minimal observable impact on tasks such as PIQA, RTE, etc. Thus, for these tasks, degradation due to KV eviction is **not “easy to notice”**.
>
> In our study, we address a more critical question: **What hidden degradation can occur when employing KV eviction, and how much eviction is acceptable before it becomes problematic?** This question is essential for understanding the practical applicability of KV cache eviction methods.
>
> Our study in Section 3 reveals that while performance on such benchmarks remains stable and appears less pronounced, KV cache eviction introduces **hidden risks** that have not been adequately revealed and addressed. In particular, we show that eviction can result in the damage of **contextual information,** which manifests as issues such as safety risks and detail hallucinations even for mild compression regimes such as 50%. Our case study (Section 3.1) is based on **actual samples** generated using KV eviction, and our controlled study on context damage is based on experiments conducted on the Line Retrieval benchmark, in which the evaluation format is widely accepted [5,6,7].
>
> Moreover, our experiments uncover significant vulnerabilities in tasks that are critical for **real-world LLM services**. As detailed in Section 5, experiments across MMLU, GSM8K, HumanEval, and Line Retrieval show rapid accuracy drops for eviction methods, emphasizing that these risks are neither theoretical nor negligible.
>
> ---
>
> > **[W3.] KV cache dropping baselines are not enough. For example, H20 is considered to be old now given the fast pace development in this field, as well as a weak baseline [1]. There are more recent KV cache dropping baselines that are strong on long-context tasks [2] [3].**
>
> In our original manuscript, we have already included **SnapKV[5] for baseline comparison**, which is a more advanced baseline compared to H2O[4]. As demonstrated in our experiments, even though SnapKV outperforms H2O, SnapKV's accuracy still drops sharply, whereas MiKV maintains high accuracy.
>
> Regarding the baselines suggested by the reviewer[2, 3], to the best of our understanding, [2] focuses on accelerating the *prefill* phase, which is orthogonal to KV cache compression and generation acceleration. As for [3], the preprint currently does not provide publicly available code, making it challenging to reproduce and compare within the given timeline. To this end, we are currently trying to implement the preprint.
>
> ---
>
> Once again, we sincerely appreciate your time and effort in reviewing our paper. If you have any remaining issues or concerns, please do not hesitate to bring them to our attention.
>
> ---
>
> **References**
>
> [1] KV Cache Compression, But What Must We Give in Return? A Comprehensive Benchmark of Long Context Capable Approaches. Yuan et al., https://arxiv.org/pdf/2407.01527
>
> [2] MInference 1.0: Accelerating Pre-filling for Long-Context LLMs via Dynamic Sparse Attention. Jiang et al., https://arxiv.org/pdf/2407.02490
>
> [3] Razorattention: Efficient kv cache compression through retrieval heads, 2024. Tang et al., https: //arxiv.org/abs/2407.15891
>
> [4] Zhang et al. “H2O: Heavy-Hitter Oracle for Efficient Generative Inference of Large Language Models”. NeurIPS 2023.
>
> [5] Li et al. “SnapKV: LLM Knows What You are Looking for Before Generation". NeurIPS 2024.
>
> [6] Jiang et al. “Mixtral of Experts". arXiv 2024.
>
> [7] Mao et al. “IceFormer: Accelerated Inference with Long-Sequence Transformers on CPUs”. ICLR 2024.

---

> > ### Comment · Reviewer_Bs5E · 2024-11-24
> >
> > Thanks the authors for the rebuttal, but my main concern on the technical novelty of the proposed method still remains. I will maintain my rating of 3 due to that reason, but lower my confidence score to 3, and let the AC decide if the paper has enough technical novelty contribution at ICLR threshold.

---

> > > ### Author Response · Authors · 2024-12-01
> > >
> > > We sincerely appreciate your additional feedback. We further address the reviewer’s concerns below.
> > >
> > > ---
> > >
> > > **[Additional Concern 1.] Thanks the authors for the rebuttal, but my main concern on the technical novelty of the proposed method still remains. I will maintain my rating of 3 due to that reason, but lower my confidence score to 3, and let the AC decide if the paper has enough technical novelty contribution at ICLR threshold.**
> > >
> > > We appreciate the reviewer’s concern regarding the technical novelty of our proposed method (MiKV) and would like to further clarify our novel contributions to address this point.
> > >
> > > The key novelty and contribution of our work lies in 1) identifying, for the first time, the practical and critical limitations of existing KV cache compression methods with concrete examples and analyses, and 2) proposing the first adaptive precision KV cache compression strategy - a practical and effective solution - to address these issues.
> > >
> > > While the proposed approach might appear simple, the underlying **technical challenges** of effectively managing KVs with irregular precision patterns in memory and accelerating subsequent matrix multiply operations are not straightforward. Thus, our **technical novelty** lies in the **design** to efficiently handle irregular quantization shapes of KVs in GPU memory and achieve practical speedups for self-attention.
> > >
> > > A key insight driving the design of MiKV is: recognizing that after positional encoding is applied, the self-attention mechanism becomes **permutation-invariant** with respect to positions in the KV cache. This means that the order of KV pairs within the cache does not affect the self-attention computation as long as KV pairs are permuted together. Leveraging this property, we proposed a **novel cache strategy** to permute and group KVs w.r.t. their precision for compression, enabling efficient management without introducing any functional consequences.
> > >
> > > First, based on the importance policy, KV pairs are partitioned and re-grouped into high-precision and low-precision groups. After grouping, each precision group is compressed with a distinct precision level. Since KVs share the same precision within each group, they can be stored contiguously in memory for efficiency. During self-attention GEMV operation, each KV group is accelerated using a INTn × FP16 GEMV kernel. Thanks to this novel cache management, MiKV is successfully **accelerated** in GPU systems. We have added more detailed explanations in **Appendix F** of the revised manuscript.
> > >
> > > To substantiate our claims, we conducted system evaluations including latency (Section 5.4) and throughput (Appendix O) analysis on NVIDIA GPUs. These experiments demonstrate that our method not only enhances the accuracy-compression tradeoff but also achieves real-world acceleration, providing **practical benefits** for LLM inference systems.
> > >
> > > In summary, our approach is not merely a straightforward integration of existing techniques but rather a carefully designed strategy that leverages importance-based adaptive precision to address the fundamental challenges of KV cache compression while achieving practical speedups.

---

> > > ### Author Response · Authors · 2024-12-01
> > > **Follow-up on additional baseline experiments ([W3])**
> > >
> > > ### **Follow-up on additional baseline experiments ([W3]).**
> > >
> > > As mentioned in our response to **W3**, our original manuscript included **SnapKV[5]** (NeurIPS 2024) as a baseline comparison, which is a recent and more advanced baseline compared to H2O[4], to ensure our method is compared against state-of-the-art methods.
> > >
> > > To further address the reviewer’s concern, we wish to add more eviction baselines. However, for the preprint kindly mentioned by the reviewer[3], it is currently challenging to re-implement the method due to the lack of available code.
> > >
> > > To this end, we conducted additional baseline experiments on two recent eviction strategy works with codes available: **kNorm[8]** (EMNLP 2024) and **TOVA[9]** (EMNLP 2024).  For Mistral-7b, we measure the GSM8K and Line Retrieval performance across three compression ratios. The results are shown in the Table below:
> > >
> > > | Method | Cache Size | GSM8K | Line Retrieval |
> > > | --- | --- | --- | --- |
> > > | **Full** | 100% | 35.2% | 100.0% |
> > > | **kNorm** [8] | 50% | 1.4% | 0.2% |
> > > |  | 25% | 1.0% | 0.0% |
> > > |  | 20% | 0.7% | 0.0% |
> > > | **TOVA** [9] | 50% | 29.3% | 64.0% |
> > > |  | 25% | 12.5% | 10.4% |
> > > |  | 20% | 8.1% | 5.4% |
> > > | **MiKV** (ours) | 50% | 35.5% | 100.0% |
> > > |  | 25% | 36.0% | 100.0% |
> > > |  | 20% | 33.4% | 97.8% |
> > >
> > > Experimental results demonstrate that the additional recent eviction-based methods also suffer from a significant drop in accuracy. In contrast, MiKV maintains high accuracy and achieves a superior accuracy-compression tradeoff. These results emphasize the common issues shared by eviction methods and highlight the robustness of our approach.
> > >
> > > ---
> > >
> > > **References**
> > >
> > > [8] Devoto et al. “A Simple and Effective L2 Norm-Based Strategy for KV Cache Compression". EMNLP 2024.
> > >
> > > [9] Oren et al. “Transformers are Multi-State RNNs”. EMNLP 2024.
> > >
> > > ---

---

### Official Review · Reviewer_BQPh · 2024-11-04

**Soundness:** 3
**Presentation:** 3
**Contribution:** 3
**Rating:** 6
**Confidence:** 4

**Summary:**

This paper investigates the drawbacks of KV cache compression methods based on eviction and quantization. It introduces the MiKV method, which applies importance-based mixed-precision compression to the KV cache, preserving less critical KV pairs in lower precision while maintaining important pairs at higher precision.

**Strengths:**

1. Proposes a mixed-precision KV cache compression approach that can be seamlessly integrated with existing importance policies.

2. Reduces quantization loss by constructing a channel balancer.

**Weaknesses:**

1. Lack of baseline comparisons: Quantization methods like KIVI are not tested on the RULER benchmark.

2. The tested context length is not specify on the  RULER benchmark, and should include additional lengths (e.g., 8k, 16k, 32k), as KV cache compression is especially beneficial at larger context lengths.

3. RULER consists mainly of synthetic tasks; it would be valuable to test on real-world tasks such as InfiniteBench or LongBench.

**Questions:**

How does MiKV handle exceptionally long contexts? Unlike methods like H2O, which can keep the KV cache size fixed, MiKV’s KV cache size will inevitably increase as the context grows.

---

> ### Author Response · Authors · 2024-11-22
> **Response to Reviewer BQPh**
>
> ---
>
> We sincerely appreciate your feedback in improving our manuscript. We address the reviewer’s concerns below.
>
> ---
>
> > **[W1.] Lack of baseline comparisons: Quantization methods like KIVI are not tested on the RULER benchmark.**
>
> Thank you for your feedback. Since our study aims to demonstrate that KV eviction poses risks for context-intensive tasks and that MiKV effectively addresses these challenges, we primarily compared our method to KV eviction(H2O[1]) to align with the message of our work. However, to address the reviewer's concerns, we conducted a comparison between KIVI[2] and our method on the RULER benchmark, which is reported below:
>
> | Method | Cache Size | wAvg Score |
> | --- | --- | --- |
> | Full | 100% | 86.0 |
> | KIVI-4 | 28% | 84.0 |
> | KIVI-2 | 17% | 61.2 |
> | H2O | 25% | 46.3 |
> | MiKV | **25%** | **85.7** |
>
> We have also added the results in **Table 3** of the revised manuscript. The experimental results show that while KIVI preserves performance in conservative precision(INT4), it lacks the flexibility to achieve varying compression rates due to its uniform-precision quantization, resulting in degradation in lower compression regimes (INT2). In contrast, MiKV, leveraging mixed-precision quantization, is capable of achieving a better accuracy-compression tradeoff. Also, note that KV eviction (H2O) suffers from performance drops due to context damage.
>
> ---
>
> > **[W2.] The tested context length is not specified on the RULER benchmark, and should include additional lengths (e.g., 8k, 16k, 32k), as KV cache compression is especially beneficial at larger context lengths.**
>
> We acknowledge the reviewer’s feedback regarding the need to test longer context lengths. To address this concern, we have conducted additional experiments on the RULER benchmark with a longer context length. Due to current resource constraints, we first experiment on 8k context length. We have revised the manuscript to include these results, which are now provided in **Appendix M**. The experimental outcomes confirm that MiKV effectively preserves accuracy even at longer context lengths.
>
> ---
>
> > **[W3.] RULER consists mainly of synthetic tasks; it would be valuable to test on real-world tasks such as InfiniteBench or LongBench.**
>
> Thank you for your suggestion. To address the reviewer’s concerns, we conducted additional experiments on the LongBench benchmark. We have revised the manuscript to include these experimental results, which are now provided in **Appendix N**. We briefly summarize the results below:
>
> | Method | Cache Size | Average |
> | --- | --- | --- |
> | Full | 100% | 46.02 |
> | KIVI-4 | 28% | 43.76 |
> | KIVI-2 | 17% | 42.88 |
> | MiKV | **25%** | **46.05** |
> | MiKV | **20%** | **45.86** |
>
> The results demonstrate similar trends with **[W1]**, in which MiKV effectively preserves performance on the LongBench dataset with better accuracy-compression tradeoff.
>
> ---
>
> > **[Q1.] How does MiKV handle exceptionally long contexts? Unlike methods like H2O, which can keep the KV cache size fixed, MiKV’s KV cache size will inevitably increase as the context grows.**
>
> As the reviewer has noted, since MiKV retains all KV pairs, the KV cache size will increase as the context length grows. In contrast, H2O can maintain a fixed KV cache size by discarding all but a predetermined number of KV pairs. However, this approach results in a significant context loss, which leads to severe accuracy drops as discussed in Sections 3 and 5 of the main manuscript. Such performance degradation negates the purpose of compression. While MiKV's KV cache size does grow with longer contexts, it ensures the preservation of context while enabling KV cache compression.
>
> ---
>
> Once again, we sincerely appreciate your time and effort in reviewing our paper. If you have any remaining issues or concerns, please do not hesitate to bring them to our attention.
>
> ---
>
> **References**
>
> [1] Zhang et al. “H2O: Heavy-Hitter Oracle for Efficient Generative Inference of Large Language Models”. NeurIPS 2023.
>
> [2] Liu et al. “KIVI: A Tuning-Free Asymmetric 2bit Quantization for KV Cache”. ICML 2024.
>
> ---

---

> > ### Comment · Reviewer_BQPh · 2024-11-24
> >
> > Thanks for the feedback. I will maintain my score.

---

> > > ### Author Response · Authors · 2024-12-02
> > >
> > > Thank you for taking the time to review our paper and response. we truly appreciate your feedback and support.

---

### Official Review · Reviewer_eieP · 2024-11-05

**Soundness:** 2
**Presentation:** 3
**Contribution:** 3
**Rating:** 5
**Confidence:** 3

**Summary:**

This work presents an efficient KV cache compression approach, which adopts adaptive quantization, assigning different elements in the KV cache bits according to their importance. This work also analyzes how missing information in the KV cache could harm the generation quality, leading to more hallucinations.

**Strengths:**

The idea is well-motivated and clear. The advantage of adaptive quantization has been widely studied in weight or activation quantization, such as AWQ. Applying a similar idea to KV cache is straightforward.

**Weaknesses:**

The novelty of this work is limited since the proposed method is more like an extension of KIVI by taking KV cache importance policies like H20 and SnapKV. One reason existing works don't use the adaptive KV cache compression is the engineering effort of implementing different quantizations with irregular shapes.

One contribution of this work could be the engineering effort or the design choice of balancing performance and efficiency. However, the implementation details are insufficient. L464 said the implementation relies on the existing kernel of KIVI. It's better to have more details or to open-source the code. Do we need triton or CUDA kernels to enable adaptive quantization?

**Questions:**

Table 3 reports the RULER benchmark, but the details of the experiments are missing. For example, the RULER has a different context length from 4K to 128K. Which length is used for the testing? Since this work said it uses Longchat, I guess the context length is only 4K, based on the result of the full KV cache setting with the RULER leaderboard.

For the latency benchmark, I am confused with the testing setting. L1409 said this work uses the Huggingface transformers library to measure the wall-clock latency, but the following content said it adopts CUDA and triton kernel for testing other models. My understanding is that the proposed method is tested with kernels but not with vLLM, sglang, or other LLM inference engines. Besides, reporting throughputs is also very helpful for testing the KV cache compression method since the size of the KV cache largely influences the parallelism of the decoding samples in a batch.

---

> ### Author Response · Authors · 2024-11-22
> **Response to Reviewer eieP (1)**
>
> We sincerely appreciate your feedback in improving our manuscript. We address the reviewer’s concerns below.
>
> ---
>
> > **[W1.]  The novelty of this work is limited since the proposed method is more like an extension of KIVI by taking KV cache importance policies like H20 and SnapKV. One reason existing works don't use the adaptive KV cache compression is the engineering effort of implementing different quantizations with irregular shapes.**
>
> **[UPDATED]** Thank you for your constructive feedback. The novelty of our work lies in two key contributions:
>
> 1. We identify and analyze previously **unrecognized hazards** exhibited by KV eviction methods, which were widely considered safe.
> 2. We introduce a **novel mixed-precision approach** to address these risks by preserving the evicted KVs in low precision and compressing important KVs in high precision.
>
> First, previous works on KV cache eviction [1, 2, 3] report seemingly minimal accuracy degradation even when evicting more than 80% of KVs, compressing the cache size down to **20%** of the original (e.g. Figure 2(d) of [1]). However, contrary to the prevailing reports that KV eviction is safe for such compression regimes, our analyses and experiments reveals that KV eviction can lead to unexpected safety risks and context damage, even under compression regimes like **50%** that were previously considered reliable (Figures 2 and 3 in the main paper). In particular, our analysis of intrinsic issues of sparsity (Section 3.2) highlights the inherent difficulty of predicting which KVs will remain important for the future, demonstrating the unreliability of eviction. To the best of our knowledge, **we are the first to reveal and analyze such hidden risks in KV cache eviction**, especially from the perspective of context damage.  By doing so, we aim to highlight scenarios where such risks (safety breaches, hallucinations, etc.) render these methods impractical from the perspective of real-world LLM services. Through evaluations across multiple benchmarks, we demonstrate how KV eviction can critically impact the reliability of LLMs in practice. Our findings emphasize the need for a more thorough evaluation of KV cache eviction methods across diverse aspects, as their risks may outweigh the benefits.
>
> Second, motivated by these findings, we propose a novel KV compression approach to mitigate the risks of KV eviction by employing mixed-precision. To the best of our knowledge, **we are the first to apply mixed-precision** quantization to LLM KV cache compression. Unlike eviction methods, our method (MiKV) recognizes that while “important” KVs are critical for LLM performance, preserving only these KVs is insufficient to ensure contextual safety. The information encoded in "less important" KVs also plays a vital role in maintaining context and must be efficiently preserved.
>
> On the other hand, other quantization approaches that use uniform precision do not take “important” KVs into consideration, resulting in significant **accuracy loss** under aggressive low-precision or **inefficient compression** rates under conservative high-precision. However, MiKV selectively compresses KVs using mixed-precision quantization based on their importance, ensuring that critical information is retained while minimizing memory usage.
>
> Also, as the reviewer pointed out, implementing mixed precision requires the engineering effort of implementing **different quantizations** with irregular shapes. Our novelty also lies in addressing this challenge effectively, designing our implementation to handle irregular quantization shapes efficiently and **achieve practical speedups**. We discuss this implementation novelty in detail in **[W2]**.
>
> In summary, our method enables an effective balance between cache compression and contextual integrity, addressing a trade-off that previous methods have overlooked.

---

> ### Author Response · Authors · 2024-11-22
> **Response to Reviewer eieP (2)**
>
> ---
>
> > **[W2.] One contribution of this work could be the engineering effort or the design choice of balancing performance and efficiency. However, the implementation details are insufficient. L464 said the implementation relies on the existing kernel of KIVI. It's better to have more details or to open-source the code. Do we need triton or CUDA general to enable adaptive quantization?**
>
> **[UPDATED]** Thank you for your valuable comment. As discussed in **[W1]**, the key contributions of our work are 1) it is the first to reveal and analyze in detail the risks and context damage associated with KV cache eviction, and 2) it is the first mixed precision KV cache compression strategy to remedy the context damage.
>
> Regarding the latter, which pertains to the reviewer’s feedback, our novelty and contribution also lie in the **implementation methodology** to handle irregular quantization shapes efficiently and achieve practical speedups. The mixed precision strategy of MiKV is implemented as follows:
>
> First, based on the importance policy, KV pairs are partitioned and re-grouped into high-precision and low-precision groups. This partitioning is feasible due to the following property: after positional encoding is applied, the self-attention mechanism becomes **permutation-invariant** with respect to positions in the KV cache. In other words, as long as KV pairs are permuted together, their order within the cache does not affect the self-attention computation. This enables arbitrary shuffling of KVs for the purpose of grouping them by precision without any functional consequences.
>
> After grouping, each precision group is compressed with a distinct precision level. Within each group, all KVs share the same precision, allowing them to be stored contiguously in memory for efficiency. During the GEMV operation in self-attention, each KV group is accelerated using a GPGPU kernel (e.g., CUDA, Triton, etc.) that executes INTn × FP16 GEMV operations. This design enables MiKV to balance compression efficiency and computational performance. We have added this detailed explanation to **Appendix F** of the revised manuscript.
>
> It is important to note that various kernel designs, such as on-the-fly dequantization (AWQ[4], KIVI[5], etc.) or lookup-based designs(LUT-GEMM[6]), can be employed to perform the necessary GEMV operations. For our experiments, we adopted an on-the-fly dequantization approach. Also, we plan to release the code in the camera-ready version.
>
> ---
>
> > **[Q1.] Table 3 reports the RULER benchmark, but the details of the experiments are missing. For example, the RULER has a different context length from 4K to 128K. Which length is used for the testing? Since this work said it uses Longchat, I guess the context length is only 4K, based on the result of the full KV cache setting with the RULER leaderboard.**
>
> Since we have used Longchat-7b, the default run setting is 4K. We have revised our manuscript (**Line 429**) to clarify this detail. Also, we have added experiments for longer sequence(8K) in **Appendix M** of our revised manuscript.

---

> ### Author Response · Authors · 2024-11-25
> **Response to Reviewer eieP (3)**
>
> ---
>
> **[Q2.] For the latency benchmark, I am confused with the testing setting. L1409 said this work uses the Huggingface transformers library to measure the wall-clock latency, but the following content said it adopts CUDA and triton kernel for testing other models. My understanding is that the proposed method is tested with kernels but not with vLLM, sglang, or other LLM inference engines. Besides, reporting throughputs is also very helpful for testing the KV cache compression method since the size of the KV cache largely influences the parallelism of the decoding samples in a batch.**
>
> **[UPDATED]** The reviewer’s understanding is correct. For the latency benchmark, we followed existing works [1, 5] and used the Huggingface framework for measurement. We modified the attention module to use custom kernels to measure the performance.
>
> Also, we thank the reviewer for the suggestion to include throughput measurements, as the KV cache size significantly influences the parallelism of decoding samples within a batch. In response, we conducted additional experiments comparing the throughput(tokens/s) of MiKV with baselines using the llama-2-7b model. Fixing the sequence length to 1024, we measure the throughput of the model across varying batch sizes of 16, 32, and 64. The results are reported in the table below:
>
> | **Method** | **Batch** |  |  |
> | --- | --- | --- | --- |
> |  | **16** | **32** | **64** |
> | Full Cache | 261.9 | 278.5 | OOM |
> | H2O (75% evicted) | 477.4 | 678.0 | 844.2 |
> | H2O (50% evicted) | 387.4 | 481.8 | 489.2 |
> | KIVI (2bit) | 334.6 | 591.4 | 819.5 |
> | KIVI (4bit) | 327.9 | 545.5 | 638.8 |
> | MiKV (avg. 3bit) | 329.4 | 575.5 | 738.0 |
>
> We also revised our manuscript to include this experiment in **Appendix O.**
>
> Experimental results demonstrate that MiKV achieves increased throughput compared to the FP16 full cache, with the improvement becoming more pronounced as the batch size increases.
>
> Unlike KIVI, which uses a uniform bitwidth quantization, MiKV employs a mixed precision strategy with an average precision of 3 bits, which balances accuracy and compression, resulting in practical speed improvements.
>
> In contrast, H2O (eviction) achieves the highest throughput by omitting computation for evicted KVs. However, this comes at the cost of significant accuracy degradation at similar compression ratios (as demonstrated in Figure 6), limiting its applicability. Even when H2O uses a conservative compression ratio (50%) to preserve accuracy, it still lags behind MiKV in both accuracy and throughput. These results highlight MiKV as a more effective solution, maintaining accuracy while optimizing throughput and memory.
>
> ---
>
> Once again, we sincerely appreciate your time and effort in reviewing our manuscript. If there are any remaining issues or concerns, please do not hesitate to inform us.
>
> ---
>
> **References**
>
> [1] Zhang et al. “H2O: Heavy-Hitter Oracle for Efficient Generative Inference of Large Language Models”. NeurIPS 2023.
>
> [2] Liu et al. “Scissorhands: Exploiting the Persistence of Importance Hypothesis for LLM KV Cache Compression at Test Time”. NeurIPS 2023.
>
> [3] Li et al. “SnapKV: LLM Knows What You are Looking for Before Generation". NeurIPS 2024.
>
> [4]: Lin et al. “AWQ: Activation-aware Weight Quantization for LLM Compression and Acceleration”. MLSys 2024.
>
> [5] Liu et al. “KIVI: A Tuning-Free Asymmetric 2bit Quantization for KV Cache”. ICML 2024.
>
> [6] Park et al. “LUT-GEMM: Quantized Matrix Multiplication based on LUTs for Efficient Inference in Large-Scale Generative Language Models”. ICLR 2024

---

> ### Author Response · Authors · 2024-11-25
> **Dear Reviewer eieP**
>
> Dear Reviewer eieP,
>
> Thank you once again for your valuable feedback. We have now completed the throughput experiments mentioned in **[Q2]** and have updated the corresponding part in our author response (above) accordingly. Additionally, we have refined our responses to **[W1]** and **[W2]** to provide more detailed explanations of our implementation and the novelty of our approach. In summary, in the author responses above, **we have updated [W1], [W2], and [Q2].**
>
> We hope these updates address your concerns and further clarify our contributions. If you have any further issues or concerns, please let us know so we can faithfully address them.
>
>
> Sincerely,
>
> Authors

---

> ### Author Response · Authors · 2024-12-02
> **Gentle Reminder**
>
> Dear Reviewer **eieP**,
>
> We sincerely appreciate the time and effort you have devoted to reviewing our paper. With **less than 24 hours** remaining until the deadline of the author-reviewer discussion period, we kindly request you to review our responses to your constructive comments.
>
> To address your constructive feedback, we conducted throughput experiments and included the results in **Appendix O**, which confirms that MiKV delivers practical throughput increase. Also, we conducted additional long context experiments and included the results in **Appendix M, N**, which further confirms the integrity of our method under long contexts.
>
> Additionally, to further verify the effectiveness of our method, we conducted benchmark experiments on Llama-2-13b, a **larger model** (we have included the results on all main benchmarks in **Appendix L**), where the results confirm that MiKV effectively outperforms existing baselines.
>
> We also conducted latency benchmarks for llama-2-13b, which confirms the practical speedup of our method scales to larger models:
>
> | Method | Model Size |  |
> | --- | --- | --- |
> |  | **7b** | **13b** |
> | Full Cache | 114.91 | OOM |
> | H2O (50%) | 66.42 | 116.65 |
> | H2O (25%) | 47.20 | 73.93 |
> | KIVI (2bit) | 54.11 | 77.34 |
> | KIVI (4bit) | 58.67 | 93.22 |
> | RTN (2bit) | 50.85 | 79.23 |
> | RTN (4bit) | 62.08 | 100.79 |
> | MiKV (avg. 3bit) | 55.60 | 84.49 |
>
> Furthermore, we conducted comparisons with **additional baseline** methods[3, 4] to further validate the effectiveness of our approach. The results are as follows:
>
> | Method | Cache Size | GSM8K | Line Retrieval |
> | --- | --- | --- | --- |
> | **Full** | 100% | 35.2% | 100.0% |
> | **kNorm [7]** | 50% | 1.4% | 0.2% |
> |  | 25% | 1.0% | 0.0% |
> |  | 20% | 0.7% | 0.0% |
> | **TOVA [8]** | 50% | 29.3% | 64.0% |
> |  | 25% | 12.5% | 10.4% |
> |  | 20% | 8.1% | 5.4% |
> | **MiKV (ours)** | 50% | 35.5% | 100.0% |
> |  | 25% | 36.0% | 100.0% |
> |  | 20% | 33.4% | 97.8% |
>
> Experimental results demonstrate that additional eviction baselines also suffer from large accuracy degradation, whereas MiKV achieves a markedly better balance between accuracy and compression, underscoring the robustness of our method.
>
> Once again, thank you for your time in reviewing our manuscript and response. If you have any remaining questions or concerns, please don’t hesitate to reach out.
>
> Sincerely,
>
> Authors
>
> ---
>
> **References**
>
> [7] Devoto et al. “A Simple and Effective L2 Norm-Based Strategy for KV Cache Compression". EMNLP 2024.
>
> [8] Oren et al. “Transformers are Multi-State RNNs”. EMNLP 2024.
>
> ---

---

> > ### Comment · Reviewer_eieP · 2024-12-02
> >
> > Thanks the authors for the detailed responses. However, I would like to keep my scores.

---

> > > ### Author Response · Authors · 2024-12-03
> > >
> > > Dear Reviewer **eieP**,
> > >
> > > Thank you for your response. In our rebuttal, we believe that we thoroughly addressed all the concerns raised in the review. However, we are currently uncertain whether our rebuttal has fully resolved your questions or if there are any remaining points that need further clarification. To this end, we kindly ask the reviewer to inform us of the details of the unaddressed points that may require additional attention so that we can provide further explanation. We would greatly appreciate your thoughts on aspects that you feel require further attention.
> > >
> > > Sincerely,
> > >
> > > Authors

---

### Meta-Review · Area_Chair_Tg7j · 2024-12-20

**Metareview:**

The paper introduces MiKV, a mixed-precision KV cache compression strategy for LLMs, optimizing memory usage while maintaining performance. Unlike traditional eviction methods that risk context loss, MiKV retains less critical KV pairs at lower precision and crucial ones at higher precision, ensuring efficiency and context integrity.

Strength:
- The idea is well-motivated, building on adaptive quantization techniques like AWQ and logically extending them to KV cache.
-  It introduces a mixed-precision KV cache compression method compatible with existing importance-based policies, effectively reducing quantization loss with a channel balancer.

Weakness:
- The main weakness lays in its technical contribution. The method is perceived as an extension of existing techniques (e.g., KIVI, SnapKV) rather than introducing groundbreaking innovations. In addition, mixed-precision quantization and token importance-based selection are already well-explored concepts.

**Additional Comments On Reviewer Discussion:**

While the authors’ rebuttal provided substantial additional experiments and clarifications, many reviewers upheld their initial ratings with concerns about limited technical novelty and incomplete experiments.

- Reviewers eieP, BQPh, and Bs5E expressed concerns about the limited technical novelty of the proposed method. While they acknowledged the authors’ detailed responses and additional experiments, they felt that these efforts did not fully address the core issues.

- Reviewers BQPh appreciated the new experiments, particularly those on long contexts and real-world benchmarks, but pointed out incomplete experimental coverage.

---

### Decision · Program_Chairs · 2025-01-22

Reject